# WHEN WILL GRADIENT METHODS CONVERGE TO MAX-MARGIN CLASSIFIER UNDER ReLU MODELS?

## ABSTRACT

We study the implicit bias of gradient descent methods in solving a binary classification problem over a linearly separable dataset. The classifier is described by a nonlinear ReLU model and the objective function adopts the exponential loss function. We first characterize the landscape of the loss function and show that there can exist spurious asymptotic local minima besides asymptotic global minima. We then show that gradient descent (GD) can converge to either a global or a local max-margin direction, or may diverge from the desired max-margin direction in a general context. For stochastic gradient descent (SGD), we show that it converges in expectation to either the global or the local max-margin direction if SGD converges. We further explore the implicit bias of these algorithms in learning a multi-neuron network under certain stationary conditions, and show that the learned classifier maximizes the margins of each sample pattern partition under the ReLU activation.

## 1 INTRODUCTION

It has been observed in various machine learning problems recently that the gradient descent (GD) algorithm and the stochastic gradient descent (SGD) algorithm converge to solutions with certain properties even without explicit regularization in the objective function. Correspondingly, theoretical analysis has been developed to explain such implicit regularization property. For example, it has been shown in Gunasekar et al. (2018; 2017) that GD converges to the solution with the minimum norm under certain initialization for regression problems, even without an explicit norm constraint.

Another type of implicit regularization, where GD converges to the max-margin classifier, has been recently studied in Gunasekar et al. (2018); Ji & Telgarsky (2018); Nacson et al. (2018a); Soudry et al. (2017; 2018) for classification problems as we describe below. Given a set of training samples $\mathbf{z}_i = (\mathbf{x}_i, y_i)$ for $i = 1, \ldots, n$, where $\mathbf{x}_i$ denotes a feature vector and $y_i \in \{-1, +1\}$ denotes the corresponding label, the goal is to find a desirable linear model (i.e., a classifier) by solving the following empirical risk minimization problem

$$\min_{\mathbf{w} \in \mathbb{R}^d} \mathcal{L}(\mathbf{w}) := \frac{1}{n} \sum_{i=1}^{n} \ell(y_i \mathbf{w}^\mathsf{T} \mathbf{x}_i). \tag{1}$$

It has been shown in Nacson et al. (2018a); Soudry et al. (2017; 2018) that if the loss function $\ell(\cdot)$ is monotonically strictly decreasing and satisfies proper tail conditions (e.g., the exponential loss), and the data are linearly separable, then GD converges to the solution $\mathbf{w}$ with infinite norm and the maximum margin direction of the data, although there is no explicit regularization towards the max-margin direction in the objective function. Such a phenomenon is referred to as the implicit bias of GD, and can help to explain some experimental results. For example, even when the training error achieves zero (i.e., the resulting model enters into the linearly separable region that correctly classifies the data), the testing error continues to decrease, because the direction of the model parameter continues to have an improved margin. Such a study has been further generalized to hold for various other types of gradient-based algorithms Gunasekar et al. (2018). Moreover, Ji & Telgarsky (2018) analyzed the convergence of GD with no assumption on the data separability, and characterized the implicit regularization to be in a subspace-based form.

The focus of this paper is on the following two fundamental issues, which have not been well addressed by existing studies.

- Existing studies so far focused only on the linear classifier model. An important question one naturally asks is what happens for the more general nonlinear leaky ReLU and ReLU models. Will GD still converge, and if so will it converge to the max-margin direction? Our study here provides new insights for the ReLU model that have not been observed for the linear model in the previous studies.

- Existing studies mainly analyzed the convergence of GD with the only exceptions Ji & Telgarsky (2018); Nacson et al. (2018b) on SGD. However, Ji & Telgarsky (2018) did not establish the convergence to the max-margin direction for SGD, and Nacson et al. (2018b) established the convergence to the max-margin solution only epochwisely for cyclic SGD (not iterationwise for SGD under random sampling with replacement). Moreover, both studies considered only the linear model. Here, our interest is to explore the iterationwise convergence of SGD under random sampling with replacement to the max-margin direction, and our result can shed insights for online SGD. Furthermore, our study provides new understanding for the nonlinear ReLU and leaky ReLU models.

## 1.1 MAIN CONTRIBUTIONS

We summarize our main contributions, where our focus is on the exponential loss function under ReLU model.

We first characterize the landscape of the empirical risk function under the ReLU model, which is nonconvex and nonsmooth. We show that such a risk function has asymptotic global minima and asymptotic spurious local minima. Such a landscape is in sharp contrast to that under the linear model previously studied in Soudry et al. (2017), where there exist only equivalent global minima.

Based on the landscape property, we show that the implicit bias property in the course of the convergence of GD can fall into four cases: converges to the asymptotic global minimum along the max-margin direction, converges to an asymptotic local minimum along a local max-margin direction, stops at a finite spurious local minimum, or oscillates between the linearly separable and misclassified regions without convergence. Such a diverse behavior is also in sharp difference from that under the linear model Soudry et al. (2017), where GD always converges to the max-margin direction.

We then take a further step to study the implicit bias of SGD. We show that the expected averaged weight vector normalized by its expected $l_2$ norm converges to the global max-margin direction or local max-margin direction, as long as SGD stays either in the linearly separable region or in a region of the local minima defined by a subset of data samples with positive label. The proof here requires considerable new technical developments, which are very different from the traditional analysis of SGD, e.g., Bottou et al. (2016); Duchi & Singer (2009); Nemirovskii et al. (1983); Shalev-Shwartz et al. (2009); Xiao (2010); Bach & Moulines (2013); Bach (2014). This is because our focus here is on the exponential loss function without attainable global/local minima, whereas traditional analysis typically assumed that the minimum of the loss function is attainable. Furthermore, our goal is to analyze the implicit bias property of SGD, which is also beyond traditional analysis of SGD.

We further extend our analysis to the leaky ReLU model and multi-neuron networks.

## 1.2 RELATED WORK

**Implicit bias of gradient descent:** Gunasekar et al. (2018) studied the implicit bias of GD and SGD for minimizing the squared loss function under bounded global minimum, and showed that some of these algorithms converge to a global minimum that is closest to the initial point. Another collection of papers Gunasekar et al. (2018); Ji & Telgarsky (2018); Nacson et al. (2018a); Soudry et al. (2017); Telgarsky (2013); Soudry et al. (2018) characterized the implicit bias of algorithms for the loss functions without attainable global minimum. Telgarsky (2013) showed that AdaBoost converges to an approximate max-margin classifier. Soudry et al. (2017; 2018) studied the convergence of GD in logistic regression with linearly separable data and showed that GD converges in direction to the solution of support vector machine at a rate of $1/\ln(t)$. Nacson et al. (2018a) improved this rate to $\ln(t)/\sqrt{t}$ under the exponential loss via normalized gradient descent. Gunasekar et al. (2018) further showed that steepest descent can lead to margin maximization under generic norms. Ji & Telgarsky (2018) analyzed the convergence of GD on an arbitrary dataset, and provided the convergence rates

along the strongly convex subspace and the separable subspace. Our work studies the convergence of GD and SGD under the nonlinear ReLU model with the exponential loss, as opposed to the linear model studied by all the above previous work on the same type of loss functions.

**Implicit bias of SGD:** Ji & Telgarsky (2018) analyzed the average SGD (under random sampling) with fixed learning rate and proved the convergence of the population risk, but did not establish the parameter convergence of SGD in the max-margin direction. Nacson et al. (2018b) established the convergence of cyclic SGD epochwisely in direction to the max-margin classifier at a rate $\mathcal{O}(1/\ln t)$. Our work differs from these two studies first in that we study the ReLU model, whereas both of these studies analyzed the linear model. Furthermore, we showed that under SGD with random sampling, the expectation of the averaged weight vector converges in direction to the max-margin classifier at a rate $\mathcal{O}(1/\sqrt{\ln t})$.

**Generalization of SGD:** There have been extensive studies of the convergence and generalization performance of SGD under various models, of which we cannot provide a comprehensive list due to the space limitations. In general, these type of studies either characterize the convergence rate of SGD or provide the generalization error bounds at the convergence of SGD, e.g., Brutzkus et al. (2017); Wang et al. (2018); Li & Liang (2018), but did not characterize the implicit regularization property of SGD, such as the convergence to the max-margin direction as provided in our paper.

## 2   ReLU Classification Model

We consider the binary classification problem, in which we are given a set of training samples $\{\mathbf{z}_1, \ldots, \mathbf{z}_n\}$. Each training sample $\mathbf{z}_i = (\mathbf{x}_i, y_i)$ contains an input data $\mathbf{x}_i$ and a corresponding binary label $y_i \in \{-1, +1\}$. We denote $I^+ := \{i : y_i = +1\}$ as the set of indices of samples with label $+1$ and denote $I^- := \{i : y_i = -1\}$ in a similar way. Their cardinalities are denoted as $n^+$ and $n^-$, respectively, and are assumed to be non-zero. We consider all datasets that are linearly separable, i.e., there exists a linear classifier $\mathbf{w}$ such that $y_i \mathbf{w}^\intercal \mathbf{x}_i > 0$ for all $i = 1, \ldots, n$.

We are interested in training a ReLU model for the classification task. In specific, for a given input data $\mathbf{x}$, the model outputs $\sigma(\mathbf{w}^\intercal \mathbf{x}_i)$, where $\sigma(v) = \max\{0, v\}$ is the ReLU activation function and $\mathbf{w}$ denotes the weight parameters. The predicted label is set to be $\mathrm{sgn}(\mathbf{w}^\intercal \mathbf{x})$. Our goal is to learn a classifier by solving the following empirical risk minimization problem, where we adopt the exponential loss.

$$\min_{\mathbf{w} \in \mathbb{R}^d} \ \mathcal{L}(\mathbf{w}) := \frac{1}{n} \sum_{i=1}^n \ell(\mathbf{w}, \mathbf{z}_i), \ \text{ where } \ \ell(\mathbf{w}, \mathbf{z}_i) = \exp(-y_i \sigma(\mathbf{w}^\intercal \mathbf{x}_i)). \tag{P}$$

The ReLU activation causes the loss function in problem (P) to be nonconvex and nonsmooth. Therefore, it is important to first understand the landscape property of the loss function, which is critical for characterizing the implicit bias property of the GD and SGD algorithms.

## 3   Implicit Bias of GD in Learning ReLU Model

### 3.1   Landscape of ReLU Model

In order to understand the convergence of GD under the ReLU model, we first study the landscape of the loss function in problem (P), which turns out to be very different from that under the linear activation model. As been shown in Soudry et al. (2017); Ji & Telgarsky (2018), the loss function in problem (P) under linear activation is convex, and achieves asymptotic global minimum, i.e., $\nabla \mathcal{L}(\alpha \mathbf{w}^*) \overset{\alpha}{\to} \mathbf{0}$ and $\mathcal{L}(\alpha \mathbf{w}^*) \overset{\alpha}{\to} 0$ as the scaling constant $\alpha \to +\infty$, only if $\mathbf{w}^*$ is in the linearly separable region. In contrast, under the ReLU model, the asymptotic critical points can be either global minimum or (spurious) local minimum depending on the training datasets, and hence the convergence property of GD can be very different in nature from that under the linear model.

The following theorem characterizes the landscape properties of problem (P). Throughout, we denote the infimum of the objective function in problem (P) as $\mathcal{L}^* = \frac{n^-}{n}$. Furthermore, we call a direction $\mathbf{w}^*$ asymptotically critical if it satisfies $\nabla \mathcal{L}(\alpha \mathbf{w}^*) \to \mathbf{0}$ as $\alpha \to +\infty$.

**Theorem 3.1** (Asymptotic landscape property). *For problem (P) under the ReLU model, any corresponding asymptotic critical direction $\mathbf{w}^*$ fall into one of the following cases:*

1. *(Asymptotic global minimum): $y_i\mathbf{w}^{*\intercal}\mathbf{x}_i > 0$ for all $i \in I^+ \cup I^-$. Then,*

$$\mathcal{L}(\alpha\mathbf{w}^*) \to \mathcal{L}^* \text{ as } \alpha \to +\infty.$$

2. *(Asymptotic local minimum): $\mathbf{w}^{*\intercal}\mathbf{x}_i > 0$ for all $i \in J^+$ and $\mathbf{w}^{*\intercal}\mathbf{x}_i \leq 0$ for all $i \in (I^+ \setminus J^+) \cup I^-$, where $J^+ \subseteq I^+$. Then,*

$$\mathcal{L}(\alpha\mathbf{w}^*) \to \mathcal{L}^* + \frac{n^+ - |J^+|}{n} \text{ as } \alpha \to +\infty.$$

3. *(Local minimum): $\mathbf{w}^{*\intercal}\mathbf{x}_i \leq 0$ for all $i \in I^+ \cup I^-$. Then,*

$$\mathcal{L}(\mathbf{w}^*) = \mathcal{L}^* + \frac{n^+}{n}.$$

To further elaborate Theorem 3.1, if $\mathbf{w}^*$ classifies all data correctly (i.e., item 1), then the objective function possibly achieves global minimum $\mathcal{L}^*$ along this direction. On the other hand, if $\mathbf{w}^*$ classifies some data with label $+1$ as $-1$ (item 2), then the objective function achieves a sub-optimal value along this direction. In the worst case where all data samples are classified as $-1$ (item 3), the ReLU unit is never activated and hence the corresponding objective function has constant value 1. We note that the cases in items 2 and 3 may or may not take place depending on specific datasets, but if they do occur, the corresponding $\mathbf{w}^*$ are spurious (asymptotic) local minima. In summary, the landscape under the ReLU model can be partitioned into different regions, where gradient descent algorithms can have different implicit bias as we show next.

## 3.2 CONVERGENCE OF GD

In this subsection, we analyze the convergence of GD in learning the ReLU model. At each iteration $t$, GD performs the update

$$\mathbf{w}_{t+1} = \mathbf{w}_t - \eta\nabla\mathcal{L}(\mathbf{w}_t), \tag{GD}$$

where $\eta$ denotes the stepsize. For the linear model whose loss function has infinitely many asymptotic global minima, it has been shown in Soudry et al. (2017) that GD always converges to the max-margin direction. Such a phenomenon is regarded as the implicit bias property of GD. Here, for the ReLU model, we are also interested in analyzing whether such an implicit-bias property still holds. Furthermore, since the loss function under the ReLU model possibly contains spurious asymptotic local minima, the convergence of GD under the ReLU model should be very different from that under the linear model.

Next, we introduce various notions of margin in order to characterize the implicit bias under the ReLU model. The global max-margin direction of samples in $I^+$ is defined as

$$\widehat{\mathbf{w}}^+ = \underset{\|\mathbf{w}\|=1}{\arg\min} \max_{i \in I^+}(\mathbf{w}^{\intercal}\mathbf{x}_i).$$

Such a notion of max-margin is natural because the ReLU activation function can suppress negative inputs. We note that here $\widehat{\mathbf{w}}^+$ may not locate in the linearly separable region, and hence it may not be parallel to any (asymptotic) global minimum. As we show next, only when $\widehat{\mathbf{w}}^+$ is in the linearly separable region, GD may converge in direction to such a max-margin direction under the ReLU model. Furthermore, for each given subset $J^+ \subseteq I^+$, we define the associated local max-margin direction $\widehat{\mathbf{w}}_J^+$ as

$$\widehat{\mathbf{w}}_J^+ = \underset{\|\mathbf{w}\|=1}{\arg\min} \max_{i \in J^+}(\mathbf{w}^{\intercal}\mathbf{x}_i).$$

We further denote the set of asymptotic local minima with respect to $J^+ \subseteq I^+$ (see Theorem 3.1 item 2) as

$$\mathcal{W}_J^+ := \{\mathbf{w}^{\intercal}\mathbf{x}_i > 0, \ \forall i \in J^+ \text{ and } \mathbf{w}^{\intercal}\mathbf{x}_i \leq 0, \ \forall i \in (I^+ \setminus J^+) \cup I^-\}.$$

Of course, $\mathcal{W}_J^+$ may or may not be empty for a certain $J^+$, and $\widehat{\mathbf{w}}_J^+$ may or may not belong to $\mathcal{W}_J^+$ depending on the specific training dataset. As we show next, only when there exists a non-empty $\mathcal{W}_J^+$ and the corresponding $\widehat{\mathbf{w}}_J^+ \in \mathcal{W}_J^+$, GD may converge to such an asymptotic local minimum $\widehat{\mathbf{w}}_J^+$ direction under the ReLU model. Next, we present the implicit bias of GD for learning the ReLU model in problem (P).

**Theorem 3.2.** *Apply GD to solve problem (P) with arbitrary initialization and a small enough constant stepsize. Then, the sequence $\{\mathbf{w}_t\}_t$ generated by GD falls into one of the following cases.*

1. $\mathcal{L}(\mathbf{w}_t) \to \mathcal{L}^*$, and $\|\frac{\mathbf{w}_t}{\|\mathbf{w}_t\|} - \widehat{\mathbf{w}}^+\| = \mathcal{O}(\frac{\ln \ln t}{\ln t})$, *where $\widehat{\mathbf{w}}^+$ is in linearly separable region;*

2. *the direction of $\mathbf{w}_t$ does not converge and oscillates between linearly separable and misclassified regions, where $\widehat{\mathbf{w}}^+$ is not in linearly separable region;*

3. $\mathcal{L}(\mathbf{w}_t) \to \mathcal{L}^* + \frac{n^+ - |J^+|}{n}$, *and* $\|\frac{\mathbf{w}_t}{\|\mathbf{w}_t\|} - \widehat{\mathbf{w}}_J^+\| = \mathcal{O}(\frac{\ln \ln t}{\ln t})$, *where $J^+ \neq \emptyset$, and $\widehat{\mathbf{w}}_J^+ \in \mathcal{W}_J^+$;*

4. $\mathcal{L}(\mathbf{w}_t) = \mathcal{L}^* + \frac{n^+}{n}$, *and $\mathbf{w}_t = \hat{\mathbf{w}}_J^+$, where $J^+ = \emptyset$, i.e., GD terminates within finite steps.*

Theorem 3.2 characterizes various instances of implicit bias of GD in learning the ReLU model, which the nature of the convergence is different from that in learning the linear model. In specific, GD can either converge in direction to the global max-margin direction $\widehat{\mathbf{w}}^+$ that leads to the global minimum, or converge to the local max-margin direction $\widehat{\mathbf{w}}_J^+$ that leads to a spurious local minimum. Furthermore, it may occur that GD oscillates between the linearly separable region and the misclassified region due to the suppression effect of ReLU function. In this case, GD does not have an implicit bias property and convergence guarantee. We provide two simple examples in the supplementary material to further elaborate these cases.

### 3.3 Implicit Bias of SGD in Learning ReLU Models

In this subsection, we analyze the convergence property and the implicit bias of SGD for solving problem (P). At each iteration $t$, SGD samples an index $\xi_t \in \{1, \ldots, n\}$ uniformly at random with replacement, and performs the update

$$\mathbf{w}_{t+1} = \mathbf{w}_t - \eta_t \nabla \ell(\mathbf{w}_t, \mathbf{z}_{\xi_t}). \tag{SGD}$$

Similarly to the convergence of GD characterized in Theorem 3.2, SGD may oscillate between the linearly separable and misclassified regions. Therefore, our major interest here is the implicit bias of SGD when it does converge either to the asymptotic global minimum or local minimum. Thus, without loss of generality, we implicitly assume that $\widehat{\mathbf{w}}^+$ is in the linearly separable region, and the relevant $\widehat{\mathbf{w}}_J^+ \in \mathcal{W}_J^+$. Otherwise, SGD does not even converge.

The implicit bias of SGD with replacement sampling has not been studied in the existing literature, and the proof of the convergence and the characterization of the implicit bias requires substantial new technical developments. In particular, traditional analysis of SGD under convex functions requires the assumption that the variance of the gradient is bounded Bottou et al. (2016); Bach (2014); Bach & Moulines (2013). Instead of making such an assumption, we next prove that SGD enjoys a nearly-constant bound on the variance up to a logarithmic factor of $t$ in learning the ReLU model.

**Proposition 1** (Variance bound). *Apply SGD to solve problem (P) with any initialization. If there exists $\mathcal{T}$ such that for all $t > \mathcal{T}$, $\mathbf{w}_t$ either stays in the linearly separable region, or in $\mathcal{W}_J^+$, then with stepsize $\eta_k = (k+1)^{-\alpha}$ where $0.5 < \alpha < 1$, the variances of the stochastic gradients sampled by SGD along the iteration path satisfy that for all $t$,*

$$\sum_{k=0}^{t-1} \eta_k^2 \mathbb{E} \|\nabla \ell(\mathbf{w}_k, \mathbf{z}_{\xi_k})\|^2 \leq \mathcal{O}\left(\frac{\ln t}{\gamma^2}\right).$$

Proposition 1 shows that the summation of the norms of the stochastic gradients grows logarithmically fast. This implies that the variance of the stochastic gradients is well-controlled. In particular, if we choose $\eta_k = (k+1)^{-1/2}$, then the bound in Proposition 1 implies that the term $\mathbb{E}\|\nabla \ell(\mathbf{w}_k, \mathbf{z}_{\xi_k})\|^2$ stays at a constant level. Based on the variance bound in Proposition 1, we next establish the convergence rate of SGD for learning the ReLU model. Throughout, we denote $\overline{\mathbf{w}}_t := \frac{1}{t} \sum_{k=0}^{t-1} \mathbf{w}_k$ as the averaged iterates generated by SGD.

**Theorem 3.3** (Convergence rate of loss). *Apply SGD to solve problem (P) with any initialization. If there exist $\mathcal{T}$ such that for all $t > \mathcal{T}$, $\mathbf{w}_t$ either stays in the linearly separable region, or in $\mathcal{W}_J^+$,*

*then with the stepsize $\eta_k = (k+1)^{-\alpha}$, where $0.5 < \alpha < 1$, the averaged iterates generated by SGD satisfies*

$$\mathbb{E}\mathcal{L}(\overline{\mathbf{w}}_t) - \mathcal{L}^* \leq \mathcal{O}\left(\frac{\ln^2 t}{t^{1-\alpha}}\right), \quad \|\mathbb{E}\overline{\mathbf{w}}_t\| \geq \mathcal{O}(\ln t).$$

*If there exist $\mathcal{T}$ such that for all $t > \mathcal{T}$, $\mathbf{w}_t$ stays in $\mathcal{W}_J^+$, then with the same stepsize*

$$\mathbb{E}\mathcal{L}(\overline{\mathbf{w}}_t) - \left(\mathcal{L}^* + \frac{n^+ - |J^+|}{n}\right) \leq \mathcal{O}\left(\frac{\ln^2 t}{t^{1-\alpha}}\right), \quad \|\mathbb{E}\overline{\mathbf{w}}_t\| \geq \mathcal{O}(\ln t).$$

Theorem 3.3 establishes the convergence rate of the expected risk of the averaged iterates generated by SGD. It can be seen that the convergence of SGD achieves different loss values corresponding to global and local minimum in different regions. The stepsize is set to be diminishing to compensate the variance introduced by SGD. In particular, if $\alpha$ is chosen to be sufficiently close to $0.5$, then the convergence rate is nearly of the order $\mathcal{O}(\ln^2 t/\sqrt{t})$, which matches the standard result of SGD in convex optimization up to an logarithmic order. Theorem 3.3 also implies that the convergence of SGD is attained as $\|\mathbb{E}\overline{\mathbf{w}}_t\| \to +\infty$ at a rate of $\mathcal{O}(\ln t)$. We note that the analysis of Theorem 3.3 is different from that of SGD in traditional convex optimization, which requires the global minimum to be achieved at a bounded point and assumes the variance of the stochastic gradients is bounded by a constant Shalev-Shwartz et al. (2009); Duchi & Singer (2009); Nemirovski et al. (2009). These assumptions do not hold here.

**Theorem 3.4** (Implicit bias of SGD). *Apply SGD to solve problem (P) with any initialization. If there exist $\mathcal{T}$ such that for all $t > \mathcal{T}$, $\mathbf{w}_t$ either stays in the linearly separable region, or in $\mathcal{W}_J^+$, then with the stepsize $\eta_k = (k+1)^{-\alpha}$ where $0.5 < \alpha < 1$, the sequence of the averaged iterate $\{\overline{\mathbf{w}}_t\}_t$ generated by SGD satisfies*

$$\left\|\frac{\mathbb{E}\overline{\mathbf{w}}_t}{\|\mathbb{E}\overline{\mathbf{w}}_t\|} - \widehat{\mathbf{w}}^+\right\|^2 = \mathcal{O}\left(\frac{1}{\ln t}\right).$$

*If there exist $\mathcal{T}$ such that for all $t > \mathcal{T}$, $\mathbf{w}_t$ stays in $\mathcal{W}_J^+$, then with the same stepsize*

$$\left\|\frac{\mathbb{E}\overline{\mathbf{w}}_t}{\|\mathbb{E}\overline{\mathbf{w}}_t\|} - \widehat{\mathbf{w}}_J^+\right\|^2 = \mathcal{O}\left(\frac{1}{\ln t}\right).$$

Theorem 3.4 shows that the direction of the expected averaged iterate $\mathbb{E}[\overline{\mathbf{w}}_t]$ generated by SGD converges to the max-margin direction $\widehat{\mathbf{w}}^+$, without any explicit regularizer in the objective function. The proof of Theorem 3.4 requires a detailed analysis of the SGD update under the ReLU model and is substantially different from that under the linear model Soudry et al. (2018); Ji & Telgarsky (2018); Nacson et al. (2018a;b). In particular, we need to handle the variance of the stochastic gradients introduced by SGD and exploit its classification properties under the ReLU model.

We next provide an example class of datasets (which has been studied in Combes et al. (2018)), for which we show that SGD stays stably in the linearly separable region.

**Proposition 2.** *If the linear separable samples $\{\mathbf{z}_1, \ldots, \mathbf{z}_n\}$ satisfy the following conditions given in Combes et al. (2018):*

*1. For all $(i,j) \in I^+ \times I^+ \cup I^- \times I^-$, it holds that $\mathbf{x}_i^\mathsf{T}\mathbf{x}_j > 0$;*
*2. For all $(i,j) \in I^+ \times I^- \cup I^- \times I^+$, it holds that $\mathbf{x}_i^\mathsf{T}\mathbf{x}_j < 0$,*

*then there exists a $\bar{t} \in \mathbb{N}$ such that for all $t \geq \bar{t}$ the sequence generated by SGD stays in the linearly separable region, as long as SGD is not initialized at the local minima described in item 3 of Theorem 3.1.*

## 4 FURTHER EXTENSIONS AND DISCUSSIONS

### 4.1 LEAKY RELU MODELS

The leaky ReLU activation takes the form $\sigma(v) = \max(\alpha v, v)$, where the parameter ($0 \leq \alpha \leq 1$). Clearly, leaky ReLU takes the linear and ReLU models as two special cases, respectively corresponding to $\alpha = 0$ and $\alpha = 1$. Since the convergence of GD/SGD of the ReLU model is very

different from that of the linear model, a natural question to ask is whether leaky ReLU with intermediate parameters $0 < \alpha < 1$ takes the same behavior as the linear or ReLU model.

It can be shown that the loss function in problem (P) under the leaky ReLU model has only asymptotic global minima achieved by $\mathbf{w}^*$ in the separable region with infinite norm (there does not exist asymptotic local minima). Hence, the convergence of GD is similar to that under the linear model, where the only difference is that the max-margin classifier needs to be defined based on leaky ReLU as follows.

For the given set of linearly separable data samples, we construct a new set of data $\mathbf{z}_i^* = (\mathbf{x}_i^*, y_i^*)$, in which $\mathbf{x}_i^* = \mathbf{x}_i, \forall i \in I^+$, $\mathbf{x}_i^* = \alpha \mathbf{x}_i, \forall i \in I^-$, and $y_i^* = y_i, \forall i \in I^+ \cup I^-$. Essentially, the data samples with label $-1$ are scaled by the parameter $\alpha$ of leaky ReLU. Without loss of generality, we assume that the max-margin classifier for data $\{\mathbf{x}_i^*\}$ passes through the origin after a proper translation. Then, we define the max-margin direction of data $\mathbf{X}^*$ as

$$\widehat{\mathbf{w}}^* = \arg \min_{\|\mathbf{w}\|=1} \max_{i \in I^+ \cup I^-} (y_i^* \mathbf{w}^\mathsf{T} \mathbf{x}_i^*).$$

Then, following the result under the linear model in Soudry et al. (2017), it can be shown that GD with arbitrary initialization and small constant stepsize for solving problem (P) under the leaky ReLU model satisfies that $\mathcal{L}(\mathbf{w})$ converges to zero, and $\mathbf{w}$ converges to the max-margin direction, i.e., $\lim_{t \to \infty} \frac{\mathbf{w}_t}{\|\mathbf{w}_t\|} = \widehat{\mathbf{w}}^*$, with its norm going to infinity.

Furthermore, following our result of Theorem 3.4, it can be shown that for SGD applied to solve problem (P) with any initialization, if there exists $\mathcal{T}$ such that for all $t > \mathcal{T}$ $\mathbf{w}_t$ stays in the linearly separable region, then with the stepsize $\eta_k = (k+1)^{-\alpha}$, $0.5 < \alpha < 1$, the sequence of the averaged iterate $\{\overline{\mathbf{w}}_t\}_t$ generated by SGD satisfies

$$\left\| \frac{\mathbb{E}\overline{\mathbf{w}}_t}{\|\mathbb{E}\overline{\mathbf{w}}_t\|} - \widehat{\mathbf{w}}^* \right\|^2 = \mathcal{O}\left( \frac{1}{\ln t} \right).$$

Thus, for SGD under the leaky ReLU model, the normalized average of the parameter vector converges in direction to the max-margin classifier.

## 4.2 Multi-neuron Networks

In this subsection, we extend our study of the ReLU model to the problem of training a one-hidden-layer ReLU neural network with $K$ hidden neurons for binary classification. Here, we do not assume linear separability of the dataset. The output of the network is given by

$$f(\mathbf{x}) = \sum_{k=1}^{K} v_k \sigma(\mathbf{w}_k^\mathsf{T} \mathbf{x}) = \mathbf{v}^\mathsf{T} \sigma(\mathbf{W}^\top \mathbf{x}), \tag{2}$$

where $\mathbf{W} = [\mathbf{w}_1, \mathbf{w}_2, \cdots, \mathbf{w}_K]$ with each column $\mathbf{w}_k$ representing the weights of the $k$th neuron in the hidden layer, $\mathbf{v}^\mathsf{T} = [v_1, v_2, \cdots, v_K]$ denotes the weights of the output neuron, and $\sigma(\cdot)$ represents the entry-wise ReLU activation function. We assume that $\mathbf{v}$ is a fixed vector whose entries are nonzero and have both positive and negative values. Such an assumption is natural as it allows the model to have enough capacity to achieve zero loss. The predicted label is set to be the sign of $f(\mathbf{x})$, and the objective function under the exponential loss is given by

$$\mathcal{L}(\mathbf{W}) = \frac{1}{n} \sum_{i=1}^{n} \exp(-y_i f(\mathbf{x}_i)). \tag{3}$$

Our goal is to characterize the implicit bias of GD and SGD for learning the weight parameters $\mathbf{W}$ of the multi-neuron model. In general, such a problem is challenging, as we have shown that GD may not converge to a desirable classifier even under the single-neuron ReLU model. For this reason, we adopt the same setting as that in (Soudry et al., 2017, Corollary 8), which assumes that the activated neurons do not change their activation status and the training error converges to zero after a sufficient number of iterations, but our result presented below characterizes the implicit bias of GD and SGD in the original feature space, which is different from that in (Soudry et al., 2017, Corollary 8). We define a set of vectors $\{\mathbf{A}_i \in \mathbb{R}^{k \times 1}\}_{i=1}^{n}$, where $\mathbf{A}_i^j = 1$ if the sample $\mathbf{x}_i$ is activated on the $j$th

neuron, i.e., $\mathbf{w}_j^\mathsf{T} \mathbf{x}_i > 0$, and set $\mathbf{A}_i^j = 0$ otherwise. Such an $\mathbf{A}_i$ vector is referred to as the activation pattern of $\mathbf{x}_i$. We then partition the set of all training samples into $m$ subsets $\mathcal{B}_1, \mathcal{B}_2, \cdots, \mathcal{B}_m$, so that the samples in the same subset have the same ReLU activation pattern, and the samples in different subsets have different ReLU activation patterns. We call $\mathcal{B}_h$, $h \in [m]$ as the $h$-th pattern partition. Let $\widetilde{\mathbf{w}}_h = \sum_{k \in \{j : \mathbf{A}_h^j = 1\}} v_k \mathbf{w}_k$. Then, for any sample $\mathbf{x} \in \mathcal{B}_h$, the output of the network is given by

$$f(\mathbf{x}) = \sum_{k=1}^{K} v_k \sigma(\mathbf{w}_k^\mathsf{T} \mathbf{x}) = \sum_{k \in \{j : \mathbf{A}_h^j = 1\}} v_k \mathbf{w}_k^\mathsf{T} \mathbf{x} = \widetilde{\mathbf{w}}_h^\mathsf{T} \mathbf{x}.$$

We next present our characterization of the implicit bias property of GD and SGD under the above ReLU network model. We define the corresponding max-margin direction of the samples in $\mathcal{B}_h$ as

$$\widehat{\mathbf{w}}_h = \arg \min_{\|\mathbf{w}\|=1} \max_{\mathbf{x} \in \mathcal{B}_h} (\mathbf{w}^\mathsf{T} \mathbf{x}).$$

Then the following theorem characterizes the implicit bias of GD under the multi-neuron network.

**Theorem 4.1.** *Suppose that GD optimizes the loss $\mathcal{L}(\mathbf{W})$ in eq. (3) to zero and there exists $\mathcal{T}$ such that for all $t > \mathcal{T}$, the neurons in the hidden layer do not change their activation status. If $\mathbf{A}_{h_1} \wedge \mathbf{A}_{h_2} = \mathbf{0}$ (where "$\wedge$" denotes the entry-wise logic operator "AND" between digits zero or one) for any $h_1 \neq h_2$, then the samples in the same pattern partition of the ReLU activation have the same label, and*

$$\left\| \frac{\widetilde{\mathbf{w}}_h^t}{\|\widetilde{\mathbf{w}}_h^t\|} - \widehat{\mathbf{w}}_h \right\| = \mathcal{O}\left(\frac{\ln \ln t}{\ln t}\right), \qquad \text{for all } h \in [m].$$

Differently from (Soudry et al., 2017, Corollary 8) which studies the convergence of the vectorized weight matrix so that the implicit bias of GD is with respect to features being lifted to an extended dimensional space, Theorem 4.1 characterizes the convergence of the weight parameters and the implicit bias in the original feature space. In particular, Theorem 4.1 implies that although the ReLU neural network is a nonlinear classifier, $f(\mathbf{x})$ is equivalent to a ReLU classifier for the samples in the same pattern partition (that are from the same class), which converges in direction to the max-margin classifier $\widehat{\mathbf{w}}_h$ of those data samples. We next let $\breve{\mathbf{w}}_h^t := \frac{1}{t} \sum_{k=0}^{t-1} \widetilde{\mathbf{w}}_h(t)$. Then the following theorem establishes the implicit bias of SGD.

**Theorem 4.2.** *Suppose that SGD optimizes the loss $\mathcal{L}(\mathbf{W})$ in eq. (3) so that there exists $\mathcal{T}$ such that for any $t > \mathcal{T}$, $\mathcal{L}(\mathbf{W}) < 1/n$, the neurons in the hidden layer do not change their activation status, and for any $h_1 \neq h_2$, $\mathbf{A}_{h_1} \wedge \mathbf{A}_{h_2} = \mathbf{0}$. Then, for the stepsize $\eta_k = (k+1)^{-\alpha}$, $0.5 < \alpha < 1$, the samples in the same pattern partition of the ReLU activation have the same label, and*

$$\left\| \frac{\mathbb{E} \breve{\mathbf{w}}_h^t}{\|\mathbb{E} \breve{\mathbf{w}}_h^t\|} - \widehat{\mathbf{w}}_h \right\|^2 = \mathcal{O}\left(\frac{1}{\ln t}\right), \qquad \text{for all } h \in [m].$$

Similarly to GD, the averaged SGD in expectation maximizes the margin for every sample partition. At the high level, Theorem 4.1 and Theorem 4.2 imply the following generalization performance of the ReLU network under study. After a sufficiently large number of iterations, the neural network partitions the data samples into different subsets, and for each subset, the distance from the samples to the decision boundary is maximized by GD and SGD. Thus, the learned classifier is robust to small perturbations of the data, resulting in good generalization performance.

## 5 CONCLUSION

In this paper, we study the problem of learning a ReLU neural network via gradient descent methods, and establish the corresponding risk and parameter convergence under the exponential loss function. In particular, we show that due to the possible existence of spurious asymptotic local minima, GD and SGD can converge either to the global or local max-margin direction, which in the nature of convergence is very different from that under the linear model in the previous studies. We also discuss the extensions of our analysis to the more general leaky ReLU model and multi-neuron networks. In the future, it is worthy to explore the implicit bias of GD and SGD in learning multi-layer neural network models and under more general (not necessarily linearly separable) datasets.

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

# Supplementary Materials

## A  PROOF OF THEOREM 3.1

The gradient $\nabla \mathcal{L}(\mathbf{w})$ is given by

$$\nabla \mathcal{L}(\mathbf{w}) = \frac{1}{n} \sum_{i=1}^{n} \nabla \ell(\mathbf{w}, \mathbf{z}_i) = -\frac{1}{n} \sum_{i=1}^{n} y_i \mathbb{1}_{\{\mathbf{w}^\mathsf{T} \mathbf{x}_i > 0\}} \exp(-y_i \mathbf{w}^\mathsf{T} \mathbf{x}_i) \mathbf{x}_i.$$

If $y_i \mathbf{w}^{*\mathsf{T}} \mathbf{x}_i \geq 0$ for all $\mathbf{x}_i \in I^+ \cup I^-$, then as $\alpha \to +\infty$, we have, ,

$$\mathcal{L}(\alpha \mathbf{w}^*) = \frac{1}{n} \sum_{i \in I^-} \exp(0) = \frac{n^-}{n} = \mathcal{L}^*,$$

and

$$\nabla \mathcal{L}(\alpha \mathbf{w}^*) = -\frac{1}{n} \sum_{i \in I^+} \exp(-\alpha \mathbf{w}^\mathsf{T} \mathbf{x}_i) \mathbf{x}_i = \mathbf{0}.$$

Recall that $J^+ \subseteq I^+$. If $\mathbf{w}^{*\mathsf{T}} \mathbf{x}_i \leq 0$ for all $i \in (I^+ \setminus J^+) \cup I^-$, then as $\alpha \to +\infty$, we obtain

$$\mathcal{L}(\alpha \mathbf{w}^*) = \frac{1}{n} \sum_{i \in (I^+ \setminus J^+) \cup I^-} \exp(0) = \frac{n^+ - |J^+| + n^-}{n} = \mathcal{L}^* + \frac{n^+ - |J^+|}{n}.$$

and

$$\nabla \mathcal{L}(\alpha \mathbf{w}^*) = -\frac{1}{n} \sum_{i \in J^+} \exp(-\alpha \mathbf{w}^\mathsf{T} \mathbf{x}_i) \mathbf{x}_i = \mathbf{0}.$$

If $\mathbf{w}^{*\mathsf{T}} \mathbf{x}_i \leq 0$ for all $i \in I^+ \cup I^-$, then

$$\mathcal{L}(\mathbf{w}^*) = \frac{1}{n} \sum_{i=1}^{n} \exp(0) = 1 = \mathcal{L}^* + \frac{n^+}{n}, \ \nabla \mathcal{L}(\mathbf{w}^*) = \mathbf{0}.$$

The proof is now complete.

## B  PROOF OF THEOREM 3.2

First consider the case when $\widehat{\mathbf{w}}^+$ is in linearly separable region and the local minimum does not exist along the updating path. We call the region where all vectors $\mathbf{w} \in \mathbb{R}^d$ satisfy $\mathbf{w}^\mathsf{T} \mathbf{x}_i < 0$ for all $i \in I^-$ as negative correctly classified region. As shown in Soudry et al. (2017), $\mathcal{L}(\mathbf{w})$ is non-negative and $L$-smooth, which implies that

$$\mathcal{L}(\mathbf{w}_{k+1}) \leq \mathcal{L}(\mathbf{w}_k) + \nabla \mathcal{L}(\mathbf{w}_k)^\mathsf{T}(\mathbf{w}_{k+1} - \mathbf{w}_k) + \frac{L}{2} \|\mathbf{w}_{k+1} - \mathbf{w}_k\|^2$$

$$= \mathcal{L}(\mathbf{w}_k) - \eta \|\nabla \mathcal{L}(\mathbf{w}_k)\|^2 + \frac{L\eta^2}{2} \|\nabla \mathcal{L}(\mathbf{w}_k)\|^2$$

$$= \mathcal{L}(\mathbf{w}_k) - \eta(1 - \frac{L\eta}{2}) \|\nabla \mathcal{L}(\mathbf{w}_k)\|^2.$$

Based on the above inequality, we have

$$\frac{\mathcal{L}(\mathbf{w}_k) - \mathcal{L}(\mathbf{w}_{k+1})}{\eta(1 - \frac{L\eta}{2})} \geq \|\nabla \mathcal{L}(\mathbf{w}_k)\|^2,$$

which, in conjunction with $0 < \eta < 2/L$, implies that

$$\sum_{k=0}^{t} \|\nabla \mathcal{L}(\mathbf{w}_k)\|^2 \leq \sum_{k=0}^{t} \frac{\mathcal{L}(\mathbf{w}_k) - \mathcal{L}(\mathbf{w}_{k+1})}{\eta(1 - \frac{L\eta}{2})} = \frac{\mathcal{L}(\mathbf{w}_0) - \mathcal{L}(\mathbf{w}_{t+1})}{\eta(1 - \frac{L\eta}{2})}.$$

Thus, we have $\|\nabla \mathcal{L}(\mathbf{w}_k)\|^2 \to 0$ as $k \to +\infty$. By Theorem 3.1, $\|\nabla \mathcal{L}(\mathbf{w}_k)\|$ vanishes only when all samples with label $-1$ are correctly classified, and thus GD enters into the negative correctly

classified region eventually and diverges to infinity. Soudry et al. (2017) Theorem 3 shows that when GD diverges to infinity, it simultaneously converges in the direction of the max-margin classifier of all samples satisfying $\mathbf{w}_t^\intercal \mathbf{x}_i > 0$. Thus, under our setting, GD either converges in the direction of the global max-margin classifer $\widehat{\mathbf{w}}^+$:

$$\left\| \frac{\mathbf{w}_t}{\|\mathbf{w}_t\|} - \widehat{\mathbf{w}}^+ \right\| = \mathcal{O}\Big( \frac{\ln \ln t}{\ln t} \Big),$$

or the local max-margin classifier $\widehat{\mathbf{w}}_J^+$:

$$\left\| \frac{\mathbf{w}_t}{\|\mathbf{w}_t\|} - \widehat{\mathbf{w}}_J^+ \right\| = \mathcal{O}\Big( \frac{\ln \ln t}{\ln t} \Big).$$

Next, consider the case when $\widehat{\mathbf{w}}^+$ is not in linearly separable region, and the local minimum does not exist along the updating path. In such a case, we conclude that GD cannot stay in the linearly separable region. Otherwise, it converges in the direction of $\widehat{\mathbf{w}}^+$ that is not in linearly separable region, which leads to a contradiction. If the asymptotic local minimum $\widehat{\mathbf{w}}_J^+$ exists, then GD may converge in its direction. If $\widehat{\mathbf{w}}_J^+$ does not exist, GD cannot stay in both the misclassified region and linearly separable region, and thus oscillates between these two regions.

In the case when GD reaches a local minimum, by Theorem 3.1, we have $\nabla \mathcal{L}(\mathbf{w}^*) = \mathbf{0}$, and thus GD stops immediately and does not diverges to infinity.

## C    EXAMPLES OF CONVERGENCE OF GD IN RELU MODEL

**Example 1** (Figure 1, left). *The dataset consists of two samples with label $+1$ and one sample with label $-1$. These samples satisfy $\mathbf{x}_1^\intercal \mathbf{x}_3 < 0$ and $\mathbf{x}_1^\intercal \mathbf{x}_2 < 0$.*

For this example, if we initialize GD at the green classifier, then GD converges to the max-margin direction of the sample $(\mathbf{x}_1, +1)$. Clearly, such a classifier misclassifies the data sample $(\mathbf{x}_2, +1)$.

**Example 2** (Figure 1, right). *The dataset consists of one sample with label $+1$ and one sample with label $-1$. These two samples satisfy $0 < \mathbf{x}_1^\intercal \mathbf{x}_2 \le 0.5\|\mathbf{x}_2\|^2$.*

For this example, if we initialize at the green classifier, then GD oscillates around the direction $\mathbf{x}_2/\|\mathbf{x}_2\|$ and does not converge.

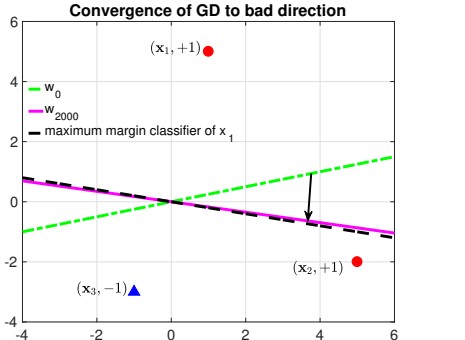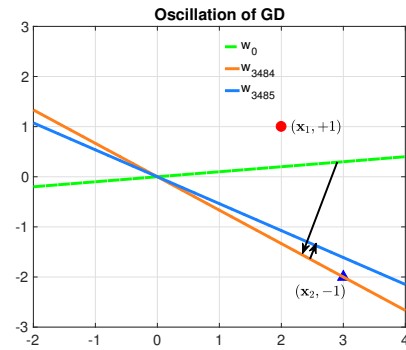

Figure 1: Failure of GD in learning ReLU models

PROOF OF EXAMPLE 1

Consider the first iteration. Note that the sample $\mathbf{z}_3$ has label $-1$, and from the illustration of Figure 1 (left) we have $\mathbf{w}_0^\intercal \mathbf{x}_3 < 0$, $\mathbf{w}_0^\intercal \mathbf{x}_2 < 0$ and $\mathbf{w}_0^\intercal \mathbf{x}_1 > 0$. Therefore, only the sample $\mathbf{z}_1$ contributes to the gradient, which is given by

$$\nabla_{\mathbf{w}_0} \mathcal{L}(\mathbf{w}_0) = -\exp(-\mathbf{w}_0^\intercal \mathbf{x}_1)\mathbf{x}_1. \tag{4}$$

By the update rule of GD, we obtain that for all $t$

$$\mathbf{w}_{t+1} = \mathbf{w}_t + \eta \exp(-\mathbf{w}_t^\mathsf{T}\mathbf{x}_1)\mathbf{x}_1. \tag{5}$$

By telescoping eq. (5), it is clear that any $\mathbf{w}_t^\mathsf{T}\mathbf{x}_2 < 0$ for all $t$ since $\mathbf{x}_1^\mathsf{T}\mathbf{x}_2 < 0$. This implies that the sample $\mathbf{z}_2$ is always misclassified.

PROOF OF EXAMPLE 2

Since we initialize GD at $\mathbf{w}_0$ such that $\mathbf{w}_0^\mathsf{T}\mathbf{x}_1 > 0$ and $\mathbf{w}_0^\mathsf{T}\mathbf{x}_2 < 0$, the sample $\mathbf{z}_2$ does not contribute to the GD update due to the ReLU activation. Next, we argue that there must exists a $t$ such that $\mathbf{w}_t^\mathsf{T}\mathbf{x}_2 > 0$. Suppose such $t$ does not exist, we always have $\mathbf{w}_t^\mathsf{T}\mathbf{x}_1 = (\mathbf{w}_0 + \sum_{k=0}^{t-1} \exp(-\mathbf{w}_k^\mathsf{T}\mathbf{x}_1)\mathbf{x}_1)^\mathsf{T}\mathbf{x}_1 > 0$. Then, the linear classifier $\mathbf{w}_t$ generated by GD stays between $\mathbf{x}_1$ and $\mathbf{x}_2$, and the corresponding objective function reduces to a linear model that depends on the sample $\mathbf{z}_1$ (Note that $\mathbf{z}_2$ contributes a constant due to ReLU activation). Following from the results in Ji & Telgarsky (2018); Soudry et al. (2017) for linear model, we conclude that $\mathbf{w}_t$ converges to the max-margin direction $\frac{\mathbf{x}_1}{\|\mathbf{x}_1\|}$ as $t \to +\infty$. Since $\mathbf{x}_1^\mathsf{T}\mathbf{x}_2 > 0$, this implies that $\mathbf{w}_t^\mathsf{T}\mathbf{x}_2 > 0$ as $t \to +\infty$, contradicting with the assumption.

Next, we consider the $t$ such that $\mathbf{w}_t^\mathsf{T}\mathbf{x}_1 > 0$ and $\mathbf{w}_t^\mathsf{T}\mathbf{x}_2 > 0$, the objective function is given by

$$\mathcal{L}(\mathbf{w}_t) = \exp(-\mathbf{w}_t^\mathsf{T}\mathbf{x}_1) + \exp(\mathbf{w}_t^\mathsf{T}\mathbf{x}_2),$$

and the corresponding gradient is given by

$$\nabla_{\mathbf{w}_t}\mathcal{L}(\mathbf{w}_t) = -\exp(-\mathbf{w}_t^\mathsf{T}\mathbf{x}_1)\mathbf{x}_1 + \exp(\mathbf{w}_t^\mathsf{T}\mathbf{x}_2)\mathbf{x}_2.$$

Next, we consider the case that $\mathbf{w}_t^\mathsf{T}\mathbf{x}_1 > 0$ for all $t$. Otherwise, both of $\mathbf{x}_1$ and $\mathbf{x}_2$ are on the negative side of the classifier and GD cannot make any progress as the corresponding gradient is zero. In the case that $\mathbf{w}_t^\mathsf{T}\mathbf{x}_1 > 0$ for all $t$, by the update rule of GD, we obtain that

$$\mathbf{w}_{t+1}^\mathsf{T}\mathbf{x}_2 - \mathbf{w}_t^\mathsf{T}\mathbf{x}_2 = \eta \exp(-\mathbf{w}_t^\mathsf{T}\mathbf{x}_1)\mathbf{x}_1^\mathsf{T}\mathbf{x}_2 - \eta \exp(\mathbf{w}_t^\mathsf{T}\mathbf{x}_2)\|\mathbf{x}_2\|^2 \le -0.5\eta\|\mathbf{x}_2\|^2. \tag{6}$$

Clearly, the sequence $\{\mathbf{w}_t^\top\mathbf{x}_2\}_t$ is strictly decreasing with a constant gap, and hence within finite steps we must have $\mathbf{w}_t^\mathsf{T}\mathbf{x}_2 \le 0$.

## D  PROOF OF PROPOSITION 1

Since SGD stays in the linearly separable region eventually, and hence only the data samples in $I^+$ contribute to the gradient update due to the ReLU activation function. For this reason, we reduce the original minimization problem (P) to the following optimization

$$\min_{\mathbf{w}\in\mathbb{R}^d} \mathcal{L}(\mathbf{w}) = \frac{1}{n^+}\sum_{i=1}^{n} \ell(\mathbf{w}, \mathbf{x}_i),$$
$$\ell(\mathbf{w}, \mathbf{x}_i) = \exp(-\mathbf{w}^\mathsf{T}\mathbf{x}_i)\mathbf{1}_{(\mathbf{x}_i\in I^+)}, \tag{7}$$

which corresponds to a linear model with samples in $I^+$. Similarly, if SGD stays in $\mathcal{W}_J^+$, only the data samples in $J^+$ contribute the the gradient update, the original minimization problem (P) is reduced to

$$\min_{\mathbf{w}\in\mathbb{R}^d} \mathcal{L}(\mathbf{w}) = \frac{1}{|J^+|}\sum_{i=1}^{n} \ell(\mathbf{w}, \mathbf{x}_i),$$
$$\ell(\mathbf{w}, \mathbf{x}_i) = \exp(-\mathbf{w}^\mathsf{T}\mathbf{x}_i)\mathbf{1}_{(\mathbf{x}_i\in J^+)}, \tag{8}$$

The proof contains three main steps.

**Step 1:** For any $\mathbf{u}$, bounding the term $\mathbb{E}\|\mathbf{w}_t - \mathbf{u}\|^2$: By the update rule of SGD, we have

$$\|\mathbf{w}_t - \mathbf{u}\|^2 = \|\mathbf{w}_{t-1} - \mathbf{u}\|^2 - 2\eta_{t-1}\langle\nabla\ell(\mathbf{w}_{t-1}, \mathbf{z}_{\xi_t}), \mathbf{w}_{t-1} - \mathbf{u}\rangle + \eta_{t-1}^2\|\nabla\ell(\mathbf{w}_{t-1}, \mathbf{z}_{\xi_t})\|^2$$
$$= \|\mathbf{w}_{t-1} - \mathbf{u}\|^2 - 2\eta_{t-1}\langle\nabla\mathcal{L}(\mathbf{w}_{t-1}), \mathbf{w}_{t-1} - \mathbf{u}\rangle + \eta_{t-1}^2\|\nabla\ell(\mathbf{w}_{t-1}, \mathbf{z}_{\xi_t})\|^2 + \mathcal{M}_t, \tag{9}$$

where
$$\mathcal{M}_t = 2\eta_{t-1}\langle\nabla\mathcal{L}(\mathbf{w}_{t-1}) - \nabla\ell(\mathbf{w}_{t-1}, \mathbf{z}_{\xi_t}), \mathbf{w}_{t-1} - \mathbf{u}\rangle.$$

By convexity we obtain that $\langle\nabla\mathcal{L}(\mathbf{w}_{t-1}), \mathbf{w}_{t-1} - \mathbf{u}\rangle \geq \mathcal{L}(\mathbf{w}_{t-1}) - \mathcal{L}(\mathbf{u})$. Then, eq. (9) further becomes

$$\|\mathbf{w}_t - \mathbf{u}\|^2 \leq \|\mathbf{w}_{t-1} - \mathbf{u}\|^2 - 2\eta_{t-1}(\mathcal{L}(\mathbf{w}_{t-1}) - \mathcal{L}(\mathbf{u})) + \eta_{t-1}^2\|\nabla\ell(\mathbf{w}_{t-1}, \mathbf{z}_{\xi_t})\|^2 + \mathcal{M}_t \quad (10)$$

Telescoping the above inequality yields that

$$\|\mathbf{w}_t - \mathbf{u}\|^2 \leq \|\mathbf{w}_0 - \mathbf{u}\|^2 - 2\sum_{k=0}^{t-1}\eta_k\mathcal{L}(\mathbf{w}_k) + 2(\sum_{k=0}^{t-1}\eta_k)\mathcal{L}(\mathbf{u}) + \sum_{k=0}^{t-1}\eta_k^2\|\nabla\ell(\mathbf{w}_k, \mathbf{z}_{\xi_k})\|^2 + \sum_{k=1}^{t}\mathcal{M}_k.$$
$$(11)$$

Taking expectation on both sides of the above inequality and note that $\mathbb{E}\mathcal{M}_t = 0$ for all $t$, we further obtain that

$$\mathbb{E}\|\mathbf{w}_t - \mathbf{u}\|^2 \leq \|\mathbf{w}_0 - \mathbf{u}\|^2 - 2\sum_{k=0}^{t-1}\eta_k\mathbb{E}\mathcal{L}(\mathbf{w}_k) + 2(\sum_{k=0}^{t-1}\eta_k)\mathcal{L}(\mathbf{u}) + \sum_{k=0}^{t-1}\eta_k^2\mathbb{E}\|\nabla\ell(\mathbf{w}_k, \mathbf{z}_{\xi_k})\|^2.$$
$$(12)$$

Note that $\ell \leq 1$ whenever the data samples are correctly classified and for all $i \in I^+$, $\|x_i\| \leq B$, and without loss of generality, we can assume $B < \sqrt{2}$. Hence, the term $\mathbb{E}\|\nabla\ell(\mathbf{w}_k, \mathbf{z}_{\xi_k})\|^2$ can be upper bounded by

$$\mathbb{E}\|\nabla\ell(\mathbf{w}_k, \mathbf{z}_{\xi_k})\|^2 = \mathbb{E}\ell(\mathbf{w}_k, \mathbf{z}_{\xi_k})\|\mathbf{z}_{\xi_k}^2\| \leq B^2\mathbb{E}\ell(\mathbf{w}_k, \mathbf{z}_{\xi_k})^2 \leq B^2\mathbb{E}\ell(\mathbf{w}_k, \mathbf{z}_{\xi_k}) = B^2\mathbb{E}\mathcal{L}(\mathbf{w}_k).$$

Then, noting that $\eta_k \leq 1$, eq. (12) can be upper bounded by

$$\mathbb{E}\|\mathbf{w}_t - \mathbf{u}\|^2 \leq \|\mathbf{w}_0 - \mathbf{u}\|^2 - (2 - B^2)\sum_{k=0}^{t-1}\eta_k\mathbb{E}\mathcal{L}(\mathbf{w}_k) + 2(\sum_{k=0}^{t-1}\eta_k)\mathcal{L}(\mathbf{u}). \quad (13)$$

Next, set $\mathbf{u} = (\ln(t)/\gamma)\hat{\mathbf{w}}_+$ and note that $\hat{\mathbf{w}}_+^\top\mathbf{x}_i \geq \gamma$ for all $i \in I^+$, we conclude that $\mathcal{L}(\mathbf{u}) = (1/n^+)\sum_{i\in I^+}\exp(-\mathbf{u}^\top\mathbf{x}_i) \leq \frac{1}{t}$. Substituting this into the above inequality and noting that $\eta_k = (k+1)^{-\alpha}$ and $0.5 < \alpha < 1$, we further obtain that

$$\mathbb{E}\|\mathbf{w}_t - \mathbf{u}\|^2 \leq \|\mathbf{w}_0\|^2 + \frac{\ln^2 t}{\gamma^2} + 2(\sum_{k=0}^{t-1}\eta_k)\frac{1}{t} - (2 - B^2)\sum_{k=0}^{t-1}\eta_k\mathbb{E}\mathcal{L}(\mathbf{w}_k)$$

$$\leq \|\mathbf{w}_0\|^2 + \frac{\ln^2 t}{\gamma^2} + \frac{2}{(1-\alpha)}t^{-\alpha} - (2 - B^2)\sum_{k=0}^{t-1}\eta_k\mathbb{E}\mathcal{L}(\mathbf{w}_k). \quad (14)$$

**Step 2:** lower bounding $\mathbb{E}\|\mathbf{w}_t - \mathbf{u}\|^2$: Note that only the samples in $I^+$ contribute to the update rule. By the update rule of SGD, we obtain that

$$\mathbf{w}_t = \mathbf{w}_0 - \sum_{k=0}^{t-1}\eta_k\nabla\ell(\mathbf{w}_k, \mathbf{z}_{\xi_k}) = \mathbf{w}_0 + \sum_{k=0}^{t-1}\eta_k\ell(\mathbf{w}_k, \mathbf{z}_{\xi_k})\mathbf{x}_{\xi_k},$$

which further implies that

$$\mathbf{w}_t^\top\hat{\mathbf{w}}_+ = \mathbf{w}_0^\top\hat{\mathbf{w}}_+ + \sum_{k=0}^{t-1}\eta_k\ell(\mathbf{w}_k, \mathbf{z}_{\xi_k})\mathbf{x}_{\xi_k}^\top\hat{\mathbf{w}}_+ \geq \mathbf{w}_0^\top\hat{\mathbf{w}}_+ + \gamma\sum_{k=0}^{t-1}\eta_k\ell(\mathbf{w}_k, \mathbf{z}_{\xi_k}).$$

Then, we can lower bound $\|\mathbf{w}_t - \mathbf{u}\|$ as

$$\|\mathbf{w}_t - \mathbf{u}\| \geq \langle\mathbf{w}_t - \mathbf{u}, \hat{\mathbf{w}}_+\rangle \geq \mathbf{w}_t^\top\hat{\mathbf{w}}_+ - \frac{\ln(t)}{\gamma}\hat{\mathbf{w}}_+^\top\hat{\mathbf{w}}_+$$

$$\geq \mathbf{w}_0^\top\hat{\mathbf{w}}_+ + \gamma\sum_{k=0}^{t-1}\eta_k\ell(\mathbf{w}_k, \mathbf{z}_{\xi_k}) - \frac{\ln(t)}{\gamma}.$$

Taking the expectation of $\|\mathbf{w}_t - \mathbf{u}\|^2$:

$$
\begin{aligned}
\mathbb{E}\|\mathbf{w}_t - \mathbf{u}\|^2 &\geq \mathbb{E}\Big(\mathbf{w}_0^\top \hat{\mathbf{w}}_+ + \gamma \sum_{k=0}^{t-1} \eta_k \ell(\mathbf{w}_k, \mathbf{z}_{\xi_k}) - \frac{\ln(t)}{\gamma}\Big)^2 \\
&\overset{(i)}{\geq} \Big(\mathbf{w}_0^\top \hat{\mathbf{w}}_+ + \gamma \sum_{k=0}^{t-1} \eta_k \mathbb{E}\ell(\mathbf{w}_k, \mathbf{z}_{\xi_k}) - \frac{\ln(t)}{\gamma}\Big)^2 \\
&= \Big(\mathbf{w}_0^\top \hat{\mathbf{w}}_+ + \gamma \sum_{k=0}^{t-1} \eta_k \mathbb{E}\mathcal{L}(\mathbf{w}_k) - \frac{\ln(t)}{\gamma}\Big)^2,
\end{aligned}
$$

where (i) follows from Jensen's inequality.

**Step 3:** Upper bounding $\sum_{k=0}^{t-1} \eta_k \mathbb{E}\mathcal{L}(\mathbf{w}_k)$: Combining the upper bound obtained in step 1 and the lower bound obtained in step 2 yields that

$$
\Big(\mathbf{w}_0^\top \hat{\mathbf{w}}_+ + \gamma \sum_{k=0}^{t-1} \eta_k \mathbb{E}\mathcal{L}(\mathbf{w}_k) - \frac{\ln(t)}{\gamma}\Big)^2 \leq \|\mathbf{w}_0\|^2 + \frac{\ln^2 t}{\gamma^2} + \frac{2}{1-\alpha} t^{-\alpha} - (2 - B^2) \sum_{k=0}^{t-1} \eta_k \mathbb{E}\mathcal{L}(\mathbf{w}_k).
$$

Solving the above quadratic inequality yields that

$$
\sum_{k=0}^{t-1} \eta_k \mathbb{E}\mathcal{L}(\mathbf{w}_k) \leq \mathcal{O}\Big(\frac{\ln t}{\gamma^2}\Big). \tag{15}
$$

## E    PROOF OF THEOREM 3.3

The proof exploits the iteration properties of SGD and the bound on the variance of SGD established in Proposition 1.

We start the proof from eq. (10), following which we obtain

$$
\mathcal{L}(\mathbf{w}_{t-1}) \leq \frac{1}{2\eta_{t-1}}(\|\mathbf{w}_{t-1} - \mathbf{u}\|^2 - \|\mathbf{w}_t - \mathbf{u}\|^2) + \mathcal{L}(\mathbf{u}) + \frac{1}{2}\eta_{t-1}\|\nabla\ell(\mathbf{w}_{t-1}, \mathbf{z}_{\xi_t})\|^2 + \frac{1}{2\eta_{t-1}}\mathcal{M}_t. \tag{16}
$$

Taking the expectation on both sides of the above inequality yields that

$$
\mathbb{E}\mathcal{L}(\mathbf{w}_{t-1}) \leq \frac{1}{2\eta_{t-1}}(\mathbb{E}\|\mathbf{w}_{t-1} - \mathbf{u}\|^2 - \mathbb{E}\|\mathbf{w}_t - \mathbf{u}\|^2) + \mathcal{L}(\mathbf{u}) + \frac{1}{2}\eta_{t-1}\mathbb{E}\|\nabla\ell(\mathbf{w}_{t-1}, \mathbf{z}_{\xi_t})\|^2,
$$

which, after telescoping, further yields that

$$
\sum_{k=0}^{t-1} \mathbb{E}\mathcal{L}(\mathbf{w}_k) \leq t\mathcal{L}(\mathbf{u}) + \frac{1}{2}\sum_{k=0}^{t-1} \frac{1}{\eta_k}(\mathbb{E}\|\mathbf{w}_k - \mathbf{u}\|^2 - \mathbb{E}\|\mathbf{w}_{k+1} - \mathbf{u}\|^2) + \frac{1}{2}\sum_{k=0}^{t-1} \eta_k \mathbb{E}\|\nabla\ell(\mathbf{w}_k, \mathbf{z}_{\xi_k})\|^2. \tag{17}
$$

By convexity of $\mathcal{L}$ in the linearly separable region, we have $\mathcal{L}\left(\frac{1}{t}\sum_{k=0}^{t-1}\mathbf{w}_k\right) \leq \frac{1}{t}\sum_{k=0}^{t-1}\mathcal{L}(\mathbf{w}_k)$, which, in conjunction with eq. (17), yields that

$$\mathbb{E}\mathcal{L}\left(\frac{1}{t}\sum_{k=0}^{t-1}\mathbf{w}_k\right)$$

$$\leq \mathcal{L}(\mathbf{u}) + \frac{1}{2t}\sum_{k=0}^{t-1}\frac{1}{\eta_k}(\mathbb{E}\|\mathbf{w}_k - \mathbf{u}\|^2 - \mathbb{E}\|\mathbf{w}_{k+1} - \mathbf{u}\|^2) + \frac{B^2}{2t}\sum_{k=0}^{t-1}\eta_k\mathbb{E}\mathcal{L}(\mathbf{w}_k)$$

$$= \mathcal{L}(\mathbf{u}) + \frac{1}{2t}\sum_{k=0}^{t-1}k^\alpha(\mathbb{E}\|\mathbf{w}_k - \mathbf{u}\|^2 - \mathbb{E}\|\mathbf{w}_{k+1} - \mathbf{u}\|^2) + \frac{B^2}{2t}\sum_{k=0}^{t-1}\eta_k\mathbb{E}\mathcal{L}(\mathbf{w}_k)$$

$$= \mathcal{L}(\mathbf{u}) + \frac{1}{2t}\sum_{k=0}^{t-1}[(k+1)^\alpha - k^\alpha]\,\mathbb{E}\|\mathbf{w}_k - \mathbf{u}\|^2 - t^\alpha\mathbb{E}\|\mathbf{w}_t - \mathbf{u}\|^2 + \frac{B^2}{2t}\sum_{k=0}^{t-1}\eta_k\mathbb{E}\mathcal{L}(\mathbf{w}_k)$$

$$\leq \mathcal{L}(\mathbf{u}) + \frac{1}{2t}\sum_{k=0}^{t-1}\frac{(k+1)^{2\alpha} - k^{2\alpha}}{(k+1)^\alpha + k^\alpha}\mathbb{E}\|\mathbf{w}_k - \mathbf{u}\|^2 + \frac{B^2}{2t}\sum_{k=0}^{t-1}\eta_k\mathbb{E}\mathcal{L}(\mathbf{w}_k)$$

$$\leq \mathcal{L}(\mathbf{u}) + \frac{1}{2t}\|\mathbf{w}_0 - \mathbf{u}\|^2 + \frac{1}{2t}\sum_{k=1}^{t-1}\frac{2\alpha(k+1)^{2\alpha-1}}{2k^\alpha}\mathbb{E}\|\mathbf{w}_k - \mathbf{u}\|^2 + \frac{B^2}{2t}\sum_{k=0}^{t-1}\eta_k\mathbb{E}\mathcal{L}(\mathbf{w}_k)$$

$$\overset{(i)}{\leq} \mathcal{L}(\mathbf{u}) + \frac{1}{2t}\|\mathbf{w}_0 - \mathbf{u}\|^2 + \frac{\alpha 4^{\alpha-1}}{t}\left(\|\mathbf{w}_0\|^2 + \frac{\ln^2 t}{\gamma^2} + \frac{2}{1-\alpha}t^{-1-\alpha}\right)\sum_{k=1}^{t-1}\frac{1}{k^{1-\alpha}} + \frac{B^2}{2t}\sum_{k=0}^{t-1}\eta_k\mathbb{E}\mathcal{L}(\mathbf{w}_k)$$

$$\leq \mathcal{L}(\mathbf{u}) + \frac{1}{2t}\|\mathbf{w}_0 - \mathbf{u}\|^2 + \frac{\alpha 4^{\alpha-1}}{t}\left(\|\mathbf{w}_0\|^2 + \frac{\ln^2 t}{\gamma^2} + \frac{2}{1-\alpha}t^{-1-\alpha}\right)\frac{t^\alpha}{\alpha} + \frac{B^2}{2t}\sum_{k=0}^{t-1}\eta_k\mathbb{E}\mathcal{L}(\mathbf{w}_k)$$

$$\leq \frac{1}{t} + \frac{1}{2t}\left(\|\mathbf{w}_0\|^2 + \frac{\ln^2 t}{\gamma^2}\right) + \frac{4^{\alpha-1}}{t^{1-\alpha}}\left(\|\mathbf{w}_0\|^2 + \frac{\ln^2 t}{\gamma^2} + \frac{2}{1-\alpha}t^{-\alpha}\right) + \frac{B^2}{2t}\sum_{k=0}^{t-1}\eta_k\mathbb{E}\mathcal{L}(\mathbf{w}_k)$$

$$= \mathcal{O}(\ln^2(t)/t^{1-\alpha})$$

where (i) follows from the fact that $\mathbb{E}\|\mathbf{w}_k - \mathbf{u}\|^2 \leq \|\mathbf{w}_0\|^2 + \frac{\ln^2 t}{\gamma^2} + \frac{2}{1-\alpha}t^{-1-\alpha}$ for $k \leq t$.

Thus, we can see that $\mathcal{L}(\overline{\mathbf{w}}_t)$ decreases to 0 at a rate of $\mathcal{O}(\ln^2(t)/t^{1-\alpha})$. If we choose $\alpha$ to be close to 0.5, the best convergence rate that can be achieved is $\mathcal{O}(\ln^2(t)/\sqrt{t})$.

## F    PROOF OF THEOREM 3.4

### F.1    MAIN PROOF OF THEOREM 3.4

We first present four technical lemmas that are useful for the proof of the main theorem.

**Lemma F.1.** *Given the stepsize $\eta_{k+1} = 1/(k+1)^{-\alpha}$ and the initialization $\mathbf{w}_{0s}$, then for $t \geq 1$, we have*

$$\|\mathbb{E}\overline{\mathbf{w}}_t\| \geq -\frac{1}{B}\ln\left(\frac{1}{t} + \frac{1}{2t}\left(\|\mathbf{w}_{0s}\|^2 + \frac{\ln^2(t)}{\gamma^2}\right)\right.$$

$$\left. + \frac{4^{\alpha-1}}{t^{1-\alpha}}\left(\|\mathbf{w}_{0s}\|^2 + \frac{\ln^2 t}{\gamma^2} + \frac{2}{1-\alpha}t^{-\alpha}\right) + \frac{B^2}{2t}\sum_{k=0}^{t-1}\eta_k\mathbb{E}\mathcal{L}(\mathbf{w}_k)\right) \tag{18}$$

**Lemma F.2.** *Let $X^+$ represent the data matrix of all samples with the label $+1$, with each row representing one sample. Then we have:*

$$\min_{\mathbf{q}\in\Delta_{n-1}}\|\mathbf{X}^{+T}\mathbf{q}\| \geq \max_{\|\mathbf{w}\|=1}\min_i(\mathbf{X}^+\mathbf{w})_i = \gamma.$$

$\Delta_{n-1}$ is the simplex in $\mathbb{R}^n$. If the equality holds (i.e., the strong duality holds) at $\overline{\mathbf{q}}$ and $\hat{\mathbf{w}}^+$, then they satisfy

$$\hat{\mathbf{w}}^+ = \frac{1}{\gamma}\mathbf{X}^+\overline{\mathbf{q}},$$

and $\hat{\mathbf{w}}^+$ is the max-margin classifier of samples with the label $+1$.

**Lemma F.3.** *Let $\gamma_p = \|\mathbb{E}\nabla\mathcal{L}(\mathbf{w}_p)\|/\mathbb{E}\mathcal{L}(\mathbf{w}_p)$ and $\hat{\eta}_{p+1} = \eta_{p+1}\mathbb{E}\mathcal{L}(\mathbf{w}_p)$ for $k \geq 1$. Then we have*

$$\ln\mathbb{E}\mathcal{L}(\mathbf{w}_k) \leq \ln\mathcal{L}(\mathbf{w}_0) - \sum_{p=0}^{k-1}\hat{\eta}_{p+1}\gamma_p\gamma + \frac{1}{2}B^4 S\sum_{p=0}^{k-1}\eta_{p+1}^2.$$

**Lemma F.4.** *For $0 \leq k \leq t-1$, we have*

$$\mathbb{E}\langle-\mathbf{w}_k, \hat{\mathbf{w}}^+\rangle \leq \frac{1}{\gamma}(\ln\mathbb{E}(\mathcal{L}(\mathbf{w}_k)) + \ln n^+).$$

We next apply the lemmas to prove the main theorem. Taking the expectation of the SGD update rule yields that

$$\mathbb{E}\mathbf{w}_k = \mathbb{E}\mathbf{w}_{k-1} - \eta_{k-1}\mathbb{E}\nabla\mathcal{L}(\mathbf{w}_{k-1}).$$

Applying the above equation recursively, we further obtain that

$$\mathbb{E}\mathbf{w}_k = \mathbf{w}_0 - \sum_{p=0}^{k-1}\eta_p\mathbb{E}\nabla\mathcal{L}(\mathbf{w}_p),$$

which further leads to

$$\sum_{k=0}^{t-1}\|\mathbb{E}\mathbf{w}_k\| \leq t\|\mathbf{w}_0\| + \sum_{k=1}^{t-1}\sum_{p=0}^{k-1}\eta_p\|\mathbb{E}\nabla\mathcal{L}(\mathbf{w}_p)\| = t\|\mathbf{w}_0\| + \sum_{p=0}^{t-2}(t-1-p)\eta_p\|\mathbb{E}\nabla\mathcal{L}(\mathbf{w}_p)\|. \quad (19)$$

Next, we prove the convergence of the direction of $\mathbb{E}[\overline{\mathbf{w}}_t]$ to the max-margin direction as follows.

$$\frac{1}{2}\left\|\frac{\mathbb{E}\overline{\mathbf{w}}_t}{\|\mathbb{E}\overline{\mathbf{w}}_t\|} - \hat{\mathbf{w}}^+\right\|^2$$

$$= 1 - \frac{\langle\mathbb{E}\overline{\mathbf{w}}_t, \hat{\mathbf{w}}^+\rangle}{\|\mathbb{E}\overline{\mathbf{w}}_t\|} = 1 + \frac{\sum_{k=0}^{t-1}\mathbb{E}\langle-\mathbf{w}_k, \hat{\mathbf{w}}^+\rangle}{t\|\mathbb{E}\overline{\mathbf{w}}_t\|}$$

$$\overset{(i)}{\leq} 1 + \sum_{k=0}^{t-1}\frac{\ln\mathbb{E}(\mathcal{L}(\mathbf{w}_k)) + \ln n^+}{\gamma t\|\mathbb{E}\overline{\mathbf{w}}_t\|}$$

$$\overset{(ii)}{\leq} 1 + \frac{\ln n^+ + \ln\mathcal{L}(\mathbf{w}_0)}{\gamma\|\mathbb{E}\overline{\mathbf{w}}_t\|} + \sum_{k=1}^{t-1}\frac{-\sum_{p=0}^{k-1}\hat{\eta}_{p+1}\gamma_p\gamma + \frac{1}{2}B^4 S\sum_{p=0}^{k-1}\eta_{p+1}^2}{\gamma t\|\mathbb{E}\overline{\mathbf{w}}_t\|}$$

$$= 1 - \sum_{k=1}^{t-1}\frac{\sum_{p=0}^{k-1}\hat{\eta}_{p+1}\gamma_p}{t\|\mathbb{E}\overline{\mathbf{w}}_t\|} + \frac{\ln n^+ + \ln\mathcal{L}(\mathbf{w}_0)}{\gamma\|\mathbb{E}\overline{\mathbf{w}}_t\|} + \frac{1}{2}B^4 S\sum_{k=1}^{t-1}\frac{\sum_{p=0}^{k-1}\eta_{p+1}^2}{\gamma t\|\mathbb{E}\overline{\mathbf{w}}_t\|}$$

$$= \frac{\left\|\sum_{k=0}^{t-1}\mathbb{E}\mathbf{w}_k\right\| - \sum_{p=0}^{t-2}(t-1-p)\eta_{p+1}\|\mathbb{E}\nabla\mathcal{L}(\mathbf{w}_p)\|}{t\|\mathbb{E}\overline{\mathbf{w}}_t\|} + \frac{\ln n^+ + \ln\mathcal{L}(\mathbf{w}_0)}{\gamma\|\mathbb{E}\overline{\mathbf{w}}_t\|} + \frac{1}{2}B^4 S\frac{\sum_{p=0}^{t-2}(t-1-p)\eta_{p+1}^2}{\gamma t\|\mathbb{E}\overline{\mathbf{w}}_t\|}$$

$$\leq \frac{\sum_{k=0}^{t-1}\|\mathbb{E}\mathbf{w}_k\| - \sum_{p=0}^{t-2}(t-1-p)\eta_{p+1}\|\mathbb{E}\nabla\mathcal{L}(\mathbf{w}_p)\|}{t\|\mathbb{E}\overline{\mathbf{w}}_t\|} + \frac{\ln n^+ + \ln\mathcal{L}(\mathbf{w}_0)}{\gamma\|\mathbb{E}\overline{\mathbf{w}}_t\|} + \frac{1}{2}B^4 S\frac{\sum_{p=0}^{t-2}\eta_{p+1}^2}{\gamma\|\mathbb{E}\overline{\mathbf{w}}_t\|}$$

$$\overset{(iii)}{\leq} \frac{\|\mathbf{w}_0\|}{\|\mathbb{E}\overline{\mathbf{w}}_t\|} + \frac{\ln n^+ + \ln\mathcal{L}(\mathbf{w}_0)}{\gamma\|\mathbb{E}\overline{\mathbf{w}}_t\|} + \frac{1}{4(\alpha - 0.5)}B^4 S\frac{1 - t^{1-2\alpha}}{\gamma\|\mathbb{E}\overline{\mathbf{w}}_t\|},$$

where (i) follows from Lemma F.4, (ii) follows from Lemma F.3 and (iii) is due to eq. (19). Since following from Lemma F.1 we have that $\|\mathbb{E}\overline{\mathbf{w}}_t\| = \mathcal{O}(\ln(t))$, the above inequality then implies that

$$\left\|\frac{\mathbb{E}\overline{\mathbf{w}}_t}{\|\mathbb{E}\overline{\mathbf{w}}_t\|} - \hat{\mathbf{w}}^+\right\|^2 = \mathcal{O}\left(\frac{1}{\ln t}\right).$$

## F.2 Proof of Technical Lemmas

*Proof of Lemma F.1.* Since $\|\mathbf{x}_i\| \leq B$ for all $i$, we obtain that

$$\exp(-\mathbb{E}\overline{\mathbf{w}}_t^T\mathbf{x}_i) \geq \exp(-\|\mathbb{E}\overline{\mathbf{w}}_t\|\|\mathbf{x}_i\|) \geq \exp(-B\|\mathbb{E}\overline{\mathbf{w}}_t\|),$$

$$\mathcal{L}(\mathbb{E}\overline{\mathbf{w}}_t) = \frac{1}{n^+}\sum_{i\in I^+}\exp(-\mathbb{E}\overline{\mathbf{w}}_t^\top\mathbf{x}_i) \geq \exp(-B\|\mathbb{E}\overline{\mathbf{w}}_t\|).$$

By convexity, we have that $\mathbb{E}\mathcal{L}(\overline{\mathbf{w}}_t) \geq \mathcal{L}(\mathbb{E}\overline{\mathbf{w}}_t)$, combining which with the above bounds further yields

$$\|\mathbb{E}\overline{\mathbf{w}}_t\| \geq -\frac{1}{B}\ln(\mathcal{L}(\mathbb{E}\overline{\mathbf{w}}_t)) \geq -\frac{1}{B}\ln(\mathbb{E}\mathcal{L}(\overline{\mathbf{w}}_t)) \geq -\frac{1}{B}\ln\left(\frac{\ln^2 t}{t^{1-\alpha}}\right).$$

That is, the increasing rate of $\mathbb{E}\|\overline{\mathbf{w}}_t\|$ is at least $\mathcal{O}(\ln(t))$. $\qquad\square$

*Proof of Lemma F.2.* Following from the definition of the max-margin, we have

$$\gamma = \max_{\|\mathbf{w}\|=1}\min_i(\mathbf{X}^+\mathbf{w})_i = \max_{\|\mathbf{w}\|\leq 1}\min_i(\mathbf{X}^+\mathbf{w})_i$$

$$= \max_{\mathbf{w}}(-\max_i(-\mathbf{X}^+\mathbf{w})_i - \mathbb{1}_{\|\mathbf{w}\|\leq 1})$$

$$= \max_{\mathbf{w}}(-f^*(-\mathbf{X}^+\mathbf{w}) - g^*(\mathbf{w}))$$

where $f^*(\mathbf{a}) = \max_i(\mathbf{a})_i$ and $g^*(\mathbf{b}) = \mathbb{1}_{\|\mathbf{b}\|\leq 1}$, and their conjugate functions are $f(\mathbf{c}) = \mathbb{1}_{\mathbf{c}\in\Delta_{n-1}}$ and $g(\mathbf{e}) = \|\mathbf{e}\|$, respectively, where $\mathbf{a}, \mathbf{b}, \mathbf{c}, \mathbf{d}$ are generic vectors. We also denote $\partial f(\mathbf{c})$ and $\partial g(\mathbf{e})$ the subgradient set of $f$ and $g$ at $\mathbf{e}$ and $\mathbf{c}$ respectively. By the Fenchel-Rockafellar duality Borwein & Lewis (2010), we obtain that

$$\gamma = \max_{\mathbf{w}} -f^*(-\mathbf{X}^+\mathbf{w}) - g(\mathbf{w}) \leq \min_{\mathbf{q}} f(\mathbf{q}) + g(\mathbf{X}^{+\top}\mathbf{q}) = \min_{\mathbf{q}\in\Delta_{n-1}}\|\mathbf{X}^{+\top}\mathbf{q}\|.$$

In particular, the strong duality holds at $\overline{\mathbf{q}}$ and $\hat{\mathbf{w}}^+$ if and only if $-\mathbf{X}^+\mathbf{w} \in \partial f(\overline{\mathbf{q}})$ and $\hat{\mathbf{w}}^+ \in \partial g(\mathbf{X}^{+\top}\overline{\mathbf{q}})$. Thus, we conclude that $\hat{\mathbf{w}}^+ = \partial g(\mathbf{X}^{+\top}\overline{\mathbf{q}}) = \frac{\mathbf{X}^{+\top}\overline{\mathbf{q}}}{\|\mathbf{X}^{+\top}\overline{\mathbf{q}}\|} = \frac{1}{\gamma}\mathbf{X}^{+\top}\overline{\mathbf{q}}.$ $\qquad\square$

*Proof of Lemma F.3.* By Taylor's expansion and the update of SGD, we obtain that

$\mathcal{L}(\mathbf{w}_k)$

$$= \mathcal{L}(\mathbf{w}_{k-1}) - \eta_k\nabla\mathcal{L}(\mathbf{w}_{k-1})^T\nabla\ell(\mathbf{w}_{k-1}, \mathbf{z}_{\xi_k}) + \frac{1}{2}\eta_k^2\nabla\ell(\mathbf{w}_{k-1}, \mathbf{z}_{\xi_k})^T\nabla^2\mathcal{L}(\widetilde{\mathbf{w}})\nabla\ell(\mathbf{w}_{k-1}, \mathbf{z}_{\xi_k}),$$

(20)

where $\widetilde{\mathbf{w}} = \theta\mathbf{w}_{k-1} + (1-\theta)\mathbf{w}_k$ for certain $0 \leq \theta \leq 1$, and is in the linear separable region. Note that for any $\mathbf{v}$,

$$\mathbf{v}^T\nabla^2\mathcal{L}(\widetilde{\mathbf{w}})\mathbf{v} = \frac{1}{n^+}\sum_{i\in I^+}\exp(-\widetilde{\mathbf{w}}^\top\mathbf{x}_i)\mathbf{v}^\top\mathbf{x}_i\mathbf{x}_i^\top\mathbf{v} \leq \frac{1}{n^+}\sum_{i\in I^+}\exp(-\widetilde{\mathbf{w}}^\top\mathbf{x}_i)\|\mathbf{x}_i\|^2\|\mathbf{v}\|^2$$

$$\leq \|\mathbf{v}\|^2 B^2 \frac{1}{n^+}\sum_{i\in I^+}\exp(-\widetilde{\mathbf{w}}^T\mathbf{x}_i) = \|\mathbf{v}\|^2 B^2\mathcal{L}(\widetilde{\mathbf{w}}) \leq \|\mathbf{v}\|^2 B^2 S,$$

where $S$ is the maximum of $\mathcal{L}(\mathbf{w})$ in the linearly separable region. We note that $S < +\infty$ because $\|\mathbf{w}\| \to \infty$ in the linearly separable region and hence $\mathcal{L}(\mathbf{w}) \to 0$. Taking the expectation on both sides of eq. (20) and recalling that

$$\mathbb{E}\|\nabla\ell(\mathbf{w}_{k-1}, \mathbf{x}_{i_k})\|^2 \leq B^2\mathbb{E}\mathcal{L}(\mathbf{w}_{k-1}),$$

we obtain that

$$\mathbb{E}\mathcal{L}(\mathbf{w}_k) \leq \mathbb{E}\mathcal{L}(\mathbf{w}_{k-1}) - \eta_k \mathbb{E}\|\nabla\mathcal{L}(\mathbf{w}_{k-1})\|^2 + \frac{1}{2}\eta_k^2 B^2 S \mathbb{E}\|\nabla\ell(\mathbf{w}_{k-1}, \mathbf{z}_{\xi_k})\|^2$$

$$\leq \mathbb{E}\mathcal{L}(\mathbf{w}_{k-1})\Big(1 - \eta_k \frac{\mathbb{E}\|\nabla\mathcal{L}(\mathbf{w}_{k-1})\|^2}{\mathbb{E}\mathcal{L}(\mathbf{w}_{k-1})} + \frac{1}{2}\eta_k^2 B^2 S \frac{\mathbb{E}\|\nabla\ell(\mathbf{w}_{k-1}, \mathbf{z}_{\xi_k})\|^2}{\mathbb{E}\mathcal{L}(\mathbf{w}_{k-1})}\Big)$$

$$\leq \mathbb{E}\mathcal{L}(\mathbf{w}_{k-1})\Big(1 - \eta_k \mathbb{E}\mathcal{L}(\mathbf{w}_{k-1}) \frac{\|\mathbb{E}\nabla\mathcal{L}(\mathbf{w}_{k-1})\|^2}{[\mathbb{E}\mathcal{L}(\mathbf{w}_{k-1})]^2} + \frac{1}{2}\eta_k^2 B^2 S \frac{\mathbb{E}\|\nabla\ell(\mathbf{w}_{k-1}, \mathbf{x}_{i_k})\|^2}{\mathbb{E}\mathcal{L}(\mathbf{w}_{k-1})}\Big)$$

$$\leq \mathbb{E}\mathcal{L}(\mathbf{w}_{k-1})(1 - \eta_k \mathbb{E}\mathcal{L}(\mathbf{w}_{k-1})\gamma_{k-1}^2 + \frac{1}{2}\eta_k^2 B^4 S).$$

Define $\hat{\eta}_k = \eta_k \mathbb{E}\mathcal{L}(\mathbf{w}_{k-1})$. Then, applying the above bound recursively yields that

$$\mathbb{E}\mathcal{L}(\mathbf{w}_k) \leq \mathcal{L}(\mathbf{w}_0) \prod_{p=0}^{k-1} (1 - \hat{\eta}_{p+1}\gamma_p^2 + \frac{1}{2}\eta_{p+1}^2 B^4 S) \tag{21}$$

$$\leq \mathcal{L}(\mathbf{w}_0) \prod_{p=0}^{k-1} \exp(-\hat{\eta}_{p+1}\gamma_p^2 + \frac{1}{2}\eta_{p+1}^2 B^4 S). \tag{22}$$

Denote $\mathbf{X} \in \mathbb{R}^{n \times d}$ as the data matrix with each row corresponding to one data sample. The derivative of the empirical risk can be written as $\nabla\mathcal{L}(\mathbf{w}) = \mathbf{X}^T \mathbf{l}(\mathbf{w})/n$, where $\mathbf{l}(\mathbf{w}) = [\ell(\mathbf{w}, \mathbf{z}_1), \ell(\mathbf{w}, \mathbf{z}_2), \dots, \ell(\mathbf{w}, \mathbf{z}_n)]$. Then, we obtain that

$$\mathbb{E}\mathcal{L}(\mathbf{w}_p) = \frac{1}{n^+} \sum_{i \in I^+} \mathbb{E}\exp(-\mathbf{w}_p^\top \mathbf{x}_i) = \frac{1}{n^+}\|\mathbb{E}(\mathbf{l}(\mathbf{w}_p))\|_1$$

and

$$\mathbb{E}\nabla\mathcal{L}(\mathbf{w}_p) = \frac{1}{n^+} \sum_{i \in I^+} \mathbb{E}\exp(-\mathbf{w}_p^\top \mathbf{x}_i)\mathbf{x}_i = \frac{1}{n^+}\mathbf{X}^+ \mathbb{E}(\mathbf{l}(\mathbf{w}_p)).$$

Based on the above relationships and Lemma F.2, we obtain that

$$\gamma_p = \frac{\|\mathbb{E}\nabla\mathcal{L}(\mathbf{w}_p)\|}{\mathbb{E}\mathcal{L}(\mathbf{w}_p)} = \frac{\|\mathbf{X}^+ \mathbb{E}(\mathbf{l}(\mathbf{w}_p))\|}{\|\mathbb{E}(\mathbf{l}(\mathbf{w}_p))\|_1} = \|\mathbf{X}^+ \frac{\mathbb{E}(\mathbf{l}(\mathbf{w}_p))}{\|\mathbb{E}(\mathbf{l}(\mathbf{w}_p))\|_1}\| \geq \gamma,$$

Taking logarithm on both sides of eq. (22) and utilizing the above facts, we further obtain that

$$\ln \mathbb{E}\mathcal{L}(\mathbf{w}_k) \leq \ln \mathcal{L}(\mathbf{w}_0) - \sum_{p=0}^{k-1} \hat{\eta}_{p+1}\gamma_p^2 + \frac{1}{2}B^4 S \sum_{p=0}^{k-1} \eta_{p+1}^2,$$

$$\leq \ln \mathcal{L}(\mathbf{w}_0) - \sum_{p=0}^{k-1} \hat{\eta}_{p+1}\gamma_p\gamma + \frac{1}{2}B^4 S \sum_{p=0}^{k-1} \eta_{p+1}^2.$$

$\square$

*Proof of Lemma F.4.* Define $h(\mathbf{y}) = \ln\left(\frac{1}{n^+}\sum_{i \in I^+} \exp(y_i)\right)$, and then its dual function $h^*(\mathbf{q}) = \ln n^+ + q_i \ln(q_i) \leq \ln n^+$. Following from Lemma F.2, $\hat{\mathbf{w}}^+ = \frac{1}{\gamma}\mathbf{X}^{+T}\overline{\mathbf{q}}$. Then, by the Fenchel-Young inequality, we obtain that

$$\mathbb{E}\langle -\mathbf{w}_k, \hat{\mathbf{w}}^+ \rangle = \frac{1}{\gamma}\langle -\mathbb{E}\mathbf{w}_k, \mathbf{X}^{+\top}\overline{\mathbf{q}} \rangle = \frac{1}{\gamma}\langle -\mathbf{X}^+ \mathbb{E}\mathbf{w}_k, \overline{\mathbf{q}} \rangle$$

$$\leq \frac{1}{\gamma}(h(-\mathbf{X}^+ \mathbb{E}\mathbf{w}_k) + h^*(\overline{\mathbf{q}})) \leq \frac{1}{\gamma}(\ln(\mathcal{L}(\mathbb{E}\mathbf{w}_k)) + \ln n^+)$$

$$\leq \frac{1}{\gamma}(\ln \mathbb{E}(\mathcal{L}(\mathbf{w}_k)) + \ln n^+).$$

$\square$

## G    PROOF OF PROPOSITION 2

Under our ReLU model, in the linearly separable region, the gradient $\nabla \mathcal{L}(\mathbf{w})$ is given by

$$\nabla \mathcal{L}(\mathbf{w}) = -\frac{1}{n} \sum_{i=1}^{n} y_i \mathbb{1}_{\{\mathbf{w}^\mathsf{T}\mathbf{x}_i > 0\}} \exp(-y_i \mathbf{w}^\mathsf{T}\mathbf{x}_i)\mathbf{x}_i = -\frac{1}{n} \sum_{i \in I^+} \exp(-\mathbf{w}^\mathsf{T}\mathbf{x}_i)\mathbf{x}_i.$$

Thus, only samples with positive classification output, i.e. $\sigma(\mathbf{w}_t^\mathsf{T}\mathbf{x}_{\xi_t}) > 0$, contribute to the SGD updates.

We first prove $\|\mathbf{w}_t\| < +\infty$ when there exist misclassified samples. Suppose, toward contradiction, that $\|\mathbf{w}_t\| = +\infty$ as $t \to +\infty$ when misclassified samples exist. Note that

$$\mathbf{w}_t = \mathbf{w}_0 + \eta \sum_{i=0}^{n} \alpha_i y_i \mathbf{x}_i \tag{23}$$

Since $\|\mathbf{w}_t\|$ is infinite, at least one of the coefficients $\alpha_i, i = 1, \cdots, n$ is infinite. No loss of generality, we assume $\alpha_p = +\infty$. Then, the inner product

$$\mathbf{w}_t^\mathsf{T}\mathbf{x}_j = \mathbf{w}_0^\mathsf{T}\mathbf{x}_j + \eta \sum_{\substack{i=0 \\ i \neq p}}^{n} \alpha_i y_i \mathbf{x}_i^\mathsf{T}\mathbf{x}_j + \alpha_p y_p \mathbf{x}_p^\mathsf{T}\mathbf{x}_j. \tag{24}$$

Based on the data selected in Proposition 2, we obtain for $\forall i \in I^- \cup I^+$

$$\forall j \in I^+, \quad y_i \mathbf{x}_i^\mathsf{T}\mathbf{x}_j > 0$$
$$\forall j \in I^-, \quad y_i \mathbf{x}_i^\mathsf{T}\mathbf{x}_j < 0,$$

which, in conjunction with eq. (24), implies that, if there exist $j \in I^+$, then the first term in the right side of eq. (24) is finite, the second term is positive, and the third term is positive and infinite. As a result, we conclude that for $\forall j \in I^+$, $\mathbf{w}^t\mathbf{x}_j > 0$ as $t \to +\infty$. Similarly, we can prove that for $\forall j \in I^-$, $\mathbf{w}^t\mathbf{x}_j \leq 0$ as $t \to +\infty$, which contracts that $\mathbf{w}^t\mathbf{x}_j > 0$. Thus, if there exist misclassified samples, then we have $\|\mathbf{w}_t\| < +\infty$.

Based on the update rule of SGD, we have, for any $j$

$$\mathbf{w}_{t+1}^\mathsf{T}\mathbf{x}_j - \mathbf{w}_t^\mathsf{T}\mathbf{x}_j = \eta \exp(-y_{\xi_t}\mathbf{w}_t^\mathsf{T}\mathbf{x}_{\xi_t})y_{\xi_t}\mathbf{x}_{\xi_t}^\mathsf{T}\mathbf{x}_j = \triangle_{\xi_t, j}. \tag{25}$$

It can be shown that

$$\forall j \in I^- \cup I^+, y_{\xi_t}\triangle_{\xi_t, j} > 0,$$

which, combined with eq. (25), implies that, if one sample is correctly classified at iteration $t$, it remains to be correctly classified in the following iterations. Next, we prove that when $\|\mathbf{w}_t\| < +\infty$, all samples are correctly classified within finite steps. Define

$$\epsilon^{++} = \min_{i_1 \in I^+, i_2 \in I^+} |\mathbf{x}_{i_1}^\mathsf{T}\mathbf{x}_{i_2}|;$$

$$\epsilon^{--} = \min_{i_1 \in I^-, i_2 \in I^-} |\mathbf{x}_{i_1}^\mathsf{T}\mathbf{x}_{i_2}|;$$

$$\epsilon^{+-} = \min_{i_1 \in I^+, i_2 \in I^-} |\mathbf{x}_{i_1}^\mathsf{T}\mathbf{x}_{i_2}|.$$

Since $\|\mathbf{w}_t\| < \infty$, there exists a constant $C$ such that $\|\mathbf{w}_t\| < C$ for all $t$. Let $D = \max_{i \in I^+} \|\mathbf{x}_i\|$. Then, we obtain, for any $j \in I^+$ and $\xi_t \in I^+$,

$$\triangle_{\xi_t, j} = \eta \exp(-\mathbf{w}_t^\mathsf{T}\mathbf{x}_{\xi_t})\mathbf{x}_{\xi_t}^\mathsf{T}\mathbf{x}_j \geq \eta \exp(-CD)\epsilon^{++},$$

and for any $j \in I^+$ and $\xi_t \in I^-$,

$$\triangle_{\xi_t, j} = -\eta \exp(\mathbf{w}_t^\mathsf{T}\mathbf{x}_{\xi_t})\mathbf{x}_{\xi_t}^\mathsf{T}\mathbf{x}_j \geq \eta \epsilon^{+-}.$$

Combining the above two inequalities $\forall j \in I^+$ yields

$$\triangle_{\xi_t, j} \geq \eta \min\{\exp(-CD)\epsilon^{++}, \eta \epsilon^{+-}\}. \tag{26}$$

Similarly, we can prove $\forall j \in I^-$

$$\triangle_{\xi_t,j} \leq -\eta \min \left\{ \exp(-CD)\epsilon^{+-}, \eta\epsilon^{--} \right\}. \tag{27}$$

Combining eq. (25), eq. (26) and eq. (27), we have, when GD is in the misclassified region, the inner product $\mathbf{w}_t^\mathsf{T}\mathbf{x}_i$ increases at least $\eta \min \left\{ \exp(-CD)\epsilon^{++}, \eta\epsilon^{+-} \right\}$ after each iteratio for $\forall \mathbf{x}_i \in I^+$ or decreases at least $\eta \min \left\{ \exp(-CD)\epsilon^{++}, \eta\epsilon^{+-} \right\}$ after each iteration for $\forall \mathbf{x}_i \in I^-$. Thus, for a sufficiently large $t$, we have

$$\forall i \in I^+, \quad \mathbf{w}_t^\mathsf{T}\mathbf{x}_i > 0$$
$$\forall i \in I^-, \quad \mathbf{w}_t^\mathsf{T}\mathbf{x}_i < 0,$$

which shows that SGD enters into linearly separable eventually. Recall that once a sample is correctly classified, it remains to be correctly classified in the following iterations. As a result, there exists $\bar{t} \in \mathbb{N}$ such that the SGD stays in linearly separable region for all $t \geq \bar{t}$.

## H    PROOF OF THEOREM 4.1

After $\mathcal{T}$ GD iterations, we randomly pick a $\mathbf{A}_r$ from $\{\mathbf{A}_i\}$. Without loss of generality, we assume that only the first $K_r$ neurons are activated for all $\mathbf{x}_i \in \mathcal{B}_r$, and suppose there are $n_r$ samples in $\mathcal{B}_r$. We first use contradiction to show that all elements in the set $\mathcal{V}_r = \{v_1, v_2, \cdots, v_{K_r}\}$ must be either all positive or all negative.

According to the update rule of GD, we have, for any $1 \leq K_1 < K_2 \leq K_r$

$$\nabla_{\mathbf{w}_{K_1}^t} \mathcal{L}(\mathbf{W}) = -\frac{v_{K_1}}{n} \sum_{\mathbf{x}_i \in \mathbf{B}_r} \exp(-y_i \widetilde{\mathbf{w}}_r^{t\mathsf{T}}\mathbf{x}_i) y_i \mathbf{x}_i,$$

$$\nabla_{\mathbf{w}_{K_2}^t} \mathcal{L}(\mathbf{W}) = -\frac{v_{K_2}}{n} \sum_{\mathbf{x}_i \in \mathbf{B}_i} \exp(-y_i \widetilde{\mathbf{w}}_r^{t\mathsf{T}}\mathbf{x}_i) y_i \mathbf{x}_i,$$

which implies that

$$\nabla_{\mathbf{w}_{K_1}^t} \mathcal{L}(\mathbf{W}) = \frac{v_{K_1}}{v_{K_2}} \nabla_{\mathbf{w}_{K_2}^t} \mathcal{L}(\mathbf{W}),$$

$$\mathbf{w}_{K_1}^{t+1} = \mathbf{w}_{K_1}^t - \eta \nabla_{\mathbf{w}_{K_1}^t} \mathcal{L}(\mathbf{W}),$$

$$\mathbf{w}_{K_2}^{t+1} = \mathbf{w}_{K_2}^t - \eta \nabla_{\mathbf{w}_{K_2}^t} \mathcal{L}(\mathbf{W}) = \mathbf{w}_{K_2}^t - \frac{v_{K_2}}{v_{K_1}} \eta \nabla_{\mathbf{w}_{K_1}^t} \mathcal{L}(\mathbf{W}). \tag{28}$$

Define the empirical risk $\mathcal{L}_r(\widetilde{\mathbf{w}}_r)$ over the samples in $\mathcal{B}_r$ as

$$\mathcal{L}_r(\widetilde{\mathbf{w}}_r^t) = \frac{1}{n_r} \sum_{x_i \in \mathcal{B}_r} \exp(-y_i f(\mathbf{x}_i)) = \frac{1}{n_r} \sum_{x_i \in \mathcal{B}_r} \exp(-y_i \widetilde{\mathbf{w}}_r^{t\mathsf{T}}\mathbf{x}_i).$$

Using that facts that $\mathcal{L}(\mathbf{W})$ converges to 0 and $\mathcal{L}_r(\widetilde{\mathbf{w}}_r^t) \leq (n/n^r)\mathcal{L}(\mathbf{W})$ implies that $\mathcal{L}_r(\widetilde{\mathbf{w}}_r^t)$ converges to 0. Thus, we have $y_i \widetilde{\mathbf{w}}_r^{t\mathsf{T}}\mathbf{x}_i \geq 0$ for all $\mathbf{x}_i \in \mathcal{B}_r$, and $\|\widetilde{\mathbf{w}}_r^t\| \rightarrow +\infty$ as $t \rightarrow +\infty$. Based on eq. (28) we have, for any $1 \leq k \leq K_r$

$$\mathbf{w}_k^t = \mathbf{w}_k^{\mathcal{T}} + \frac{v_k}{v_1} \Delta \mathbf{w}^t. \tag{29}$$

Recalling that $\widetilde{\mathbf{w}}_r^t = \sum_{k=1}^{K_r} v_k \mathbf{w}_k^t$, we rewrite $\widetilde{\mathbf{w}}_r^t$ as

$$\widetilde{\mathbf{w}}_r^t = \sum_{k=1}^{K_r} v_k \mathbf{w}_k^{\mathcal{T}} + \left( \sum_{k=1}^{K_r} \frac{v_k^2}{v_1} \right) \Delta \mathbf{w}^t \tag{30}$$

Noting that the norm of the first term in the right side of eq. (30) is finite and recalling that $\|\widetilde{\mathbf{w}}_r^t\| = +\infty$, we have, $\|\Delta \mathbf{w}^t\| = +\infty$, which, in conjunction with eq. (29), implies that $\|\mathbf{w}_k^t\| = +\infty$ for all $1 \leq k \leq K_r$.

Next, let us look at the update of $\mathbf{w}_1$ and $\mathbf{w}_2$. If there exist two elements in $\mathcal{V}_r$ that have different signs (without loss of generality, we assume $v_1 > 0$ and $v_2 < 0$), then

$$\mathbf{w}_1^{t+1} = \mathbf{w}_1^t - \eta \nabla_{\mathbf{w}_1^t} \mathcal{L}(\mathbf{W}), \tag{31}$$

$$\mathbf{w}_2^{t+1} = \mathbf{w}_2^t - \eta \nabla_{\mathbf{w}_2^t} \mathcal{L}(\mathbf{W}) = \mathbf{w}_1^t + \eta \left| \frac{v_1}{v_2} \right| \nabla_{\mathbf{w}_1^t} \mathcal{L}(\mathbf{W}) \tag{32}$$

which can be rewriten as

$$\mathbf{w}_1^t = \mathbf{w}_1^{\mathcal{T}} - \sum_{s=\mathcal{T}+1}^{t-1} \eta \nabla_{\mathbf{w}_1^t} \mathcal{L}(\mathbf{W}) = \mathbf{w}_1^{\mathcal{T}} + \Delta \mathbf{w}^t,$$

$$\mathbf{w}_2^t = \mathbf{w}_2^{\mathcal{T}} - \left|\frac{v_1}{v_2}\right| \Delta \mathbf{w}^t.$$

Recalling that $\|\Delta \mathbf{w}^t\| = +\infty$ as $t \to +\infty$ and noting that the first two neurons are activated after $\mathcal{T}$ iterations, we have, for $\forall \mathbf{x}_i \in \mathcal{B}_r$

$$\mathbf{w}_1^{t\mathsf{T}} \mathbf{x}_i = \mathbf{w}_1^{\mathcal{T}\mathsf{T}} \mathbf{x}_i + \Delta \mathbf{w}^{t\mathsf{T}} \mathbf{x}_i > 0,$$

$$\mathbf{w}_2^{t\mathsf{T}} \mathbf{x}_i = \mathbf{w}_2^{\mathcal{T}\mathsf{T}} \mathbf{x}_i - \frac{v_1}{v_2} \Delta \mathbf{w}^{t\mathsf{T}} \mathbf{x}_i > 0. \tag{33}$$

Since $\Delta \mathbf{w}^t$ belongs to the space spanned by the samples in $\mathcal{B}_r$, $\Delta \mathbf{w}^t$ cannot be perpendicular to all $\mathbf{x}_i \in \mathcal{B}_r$. Thus, when $t \to +\infty$, we can find a sample $\mathbf{x}_r \in \mathcal{B}_r$ such that $\|\Delta \mathbf{w}^{t\mathsf{T}} \mathbf{x}_r\| = +\infty$. If $\Delta \mathbf{w}^{t\mathsf{T}} \mathbf{x}_r > 0$, then we have $\mathbf{w}_2^{t\mathsf{T}} \mathbf{x}_r < 0$, which contradicts eq. (33). If $\Delta \mathbf{w}^{t\mathsf{T}} \mathbf{x}_r < 0$, then $\mathbf{w}_1^{t\mathsf{T}} \mathbf{x}_r < 0$, which also contradicts eq. (33). As a result, all elements in $\mathcal{V}_r$ have the same sign.

Next, we prove that all samples in the same pattern partition have the same label. First consider the case when all elements in $\mathcal{V}_r$ are positive. If there exists a sample $\mathbf{x}_{sp} \in \mathcal{B}_r$ such that $y_{sp} = -1$ when $t = +\infty$, then

$$\mathcal{L}(\mathbf{W}) = \frac{1}{n} \sum_{i=1}^{n} \exp(-y_i f(\mathbf{x}_i)) > \frac{1}{n} \exp(-y_{sp} \widetilde{\mathbf{w}}_r^{t\mathsf{T}} \mathbf{x}_{sp}) = \frac{1}{n} \exp\left(\sum_{k=1}^{K_r} v_k \mathbf{w}_k^{t\mathsf{T}} \mathbf{x}_{sp}\right) > \frac{1}{n},$$

which contradicts that $\mathcal{L}(\mathbf{W})$ converges to $0$.

Next, consider the case when all elements in $\mathcal{V}_r$ are negative. If there exists a sample $\mathbf{x}_{sp} \in \mathcal{B}_r$ such that $y_{sp} = +1$, we have

$$\mathcal{L}(\mathbf{W}) = \frac{1}{n} \sum_{i=1}^{n} \exp(-y_i f(\mathbf{x}_i)) > \frac{1}{n} \exp(-y_{sp} \widetilde{\mathbf{w}}_r^{t\mathsf{T}} \mathbf{x}_{sp}) = \frac{1}{n} \exp(-\sum_{k=1}^{K_r} v_k \mathbf{w}_k^{t\mathsf{T}} \mathbf{x}_{sp}) > \frac{1}{n},$$

which also leads a contradiction. Combining these two results, we conclude that if all elements in $\mathcal{V}_r$ are positive, then all samples in $\mathcal{B}_r$ have label $+1$, and if all elements in $\mathcal{V}_r$ are negative, then all samples in $\mathcal{B}_r$ have label $-1$.

Finally, note that $\widetilde{\mathbf{w}}_r$ is updated by

$$\widetilde{\mathbf{w}}_r^{t+1} = \widetilde{\mathbf{w}}_r^t + \eta \sum_{k=1}^{K_r} v_k \nabla_{\mathbf{w}_k^t} \mathcal{L}(\mathbf{W}) = \widetilde{\mathbf{w}}_r^t + \frac{\eta}{n} \left(\sum_{k=1}^{K_r} v_k^2\right) \sum_{\mathbf{x}_i \in \mathbf{B}_r} \exp(-y_i \widetilde{\mathbf{w}}_r^{t\mathsf{T}} \mathbf{x}_i) y_i \mathbf{x}_i,$$

which can be rewritten as

$$\widetilde{\mathbf{w}}_r^{t+1} = \widetilde{\mathbf{w}}_r^t + \eta \frac{n_r}{n} \left(\sum_{k=1}^{K_r} v_k^2\right) \nabla \mathcal{L}_r(\widetilde{\mathbf{w}}_r^t). \tag{34}$$

Applying Theorem 3.2 to eq. (34) with stepsize $\hat{\eta} = \eta n / \left(n_r \sum_{k=1}^{K_r} v_k^2\right)$, we obtain that $\widetilde{\mathbf{w}}_r^t$ converges in the direction of the max-marigin classifier over all samples in $\mathcal{B}_r$, i.e.,

$$\left\| \frac{\widetilde{\mathbf{w}}_r^t}{\|\widetilde{\mathbf{w}}_r^t\|} - \widehat{\mathbf{w}}_r \right\| = \mathcal{O}\left(\frac{\ln \ln t}{\ln t}\right).$$

## I   PROOF OF THEOREM 4.2

After $\mathcal{T}$ GD iterations, we randomly pick a $\mathbf{A}_r$ from $\{\mathbf{A}_i\}$. Without loss of generality, we assume that only the first $K_r$ neurons are activated for all $\mathbf{x}_i \in \mathcal{B}_r$, and suppose there are $n_r$ samples in $\mathcal{B}_r$. We first use contradiction to show that all elements in the set $\mathcal{V}_r = \{v_1, v_2, \cdots, v_{K_r}\}$ must be either all positive or all negative.

According to the update rule of SGD, for any $1 \leq K_1 < K_2 \leq K_r$

$$\nabla_{\mathbf{w}_{K_1}^t} \ell(\mathbf{W}) = -v_{K_1} \exp(-y_{\xi_t} \widetilde{\mathbf{w}}_r^{t\mathsf{T}} \mathbf{x}_{\xi_t}) y_{\xi_t} \mathbf{x}_{\xi_t},$$

$$\nabla_{\mathbf{w}_{K_2}^t} \ell(\mathbf{W}) = -v_{K_2} \exp(-y_{\xi_t} \widetilde{\mathbf{w}}_r^{t\mathsf{T}} \mathbf{x}_{\xi_t}) y_{\xi_t} \mathbf{x}_{\xi_t},$$

which implies that

$$\nabla_{\mathbf{w}_{K_1}^t} \ell(\mathbf{W}) = \frac{v_{K_1}}{v_{K_2}} \nabla_{\mathbf{w}_{K_2}^t} \ell(\mathbf{W}),$$

$$\mathbf{w}_{K_1}^{t+1} = \mathbf{w}_{K_1}^t - \eta_t \nabla_{\mathbf{w}_{K_1}^t} \ell(\mathbf{W}),$$

$$\mathbf{w}_{K_2}^{t+1} = \mathbf{w}_{K_2}^t - \eta_t \nabla_{\mathbf{w}_{K_2}^t} \ell(\mathbf{W}) = \mathbf{w}_{K_2}^t - \frac{v_{K_2}}{v_{K_1}} \eta_t \nabla_{\mathbf{w}_{K_1}^t} \ell(\mathbf{W}). \tag{35}$$

Then we prove $\widetilde{\mathbf{w}}_r^t$ diverges to infinity as $t \to +\infty$. If $\widetilde{\mathbf{w}}_r^t$ does not diverges to infinity, then there exist a positive constant $F < +\infty$ such that $\forall t \geq 0, \|\widetilde{\mathbf{w}}_r^t\| < F$. According to the update rule of SGD

$$\widetilde{\mathbf{w}}_r^{t+1} = \widetilde{\mathbf{w}}_r^t + \eta_t \Big( \sum_{k=1}^{K_r} v_k^2 \Big) \exp(-y_{\xi_t} \widetilde{\mathbf{w}}_r^{t\mathsf{T}} \mathbf{x}_{\xi_t}) y_{\xi_t} \mathbf{x}_{\xi_t}.$$

For all $\mathbf{x}_{\xi_t} \in \mathcal{B}_r$, we have $y_{\xi_t} \widetilde{\mathbf{w}}_r^{t\mathsf{T}} \mathbf{x}_{\xi_t} > 0$, thus $\|\widetilde{\mathbf{w}}_r^t\|$ is strictly increasing at each step. Since $\|\widetilde{\mathbf{w}}_r^t\|$ is upper bounded by $F$ and is in the linearly separable region, we can find a constant $\epsilon_r > 0$ such that

$$\|\widetilde{\mathbf{w}}_r^{t+1}\| \geq \|\widetilde{\mathbf{w}}_r^t\| + \eta_t \epsilon_r.$$

Recall $\eta_t = 1/(t+1)^{-\alpha}$, telescoping the above inequality from step $\mathcal{T}$ to $t = +\infty$

$$\|\widetilde{\mathbf{w}}_r^t\| \geq \|\widetilde{\mathbf{w}}_r^{\mathcal{T}}\| + \epsilon_r \sum_{s=\mathcal{T}+1}^{t-1} \frac{1}{(1+s)^\alpha},$$

since $0.5 < \alpha < 1$, the R.H.S of the above inequation goes to infinity, thus $\|\widetilde{\mathbf{w}}_r^t\| = +\infty$ when $t \to +\infty$, which is a contradiction. Thus, $\widetilde{\mathbf{w}}_r^t$ diverges to infinity.

Based on eq. (28) we have, for any $1 \leq k \leq K_r$

$$\mathbf{w}_k^t = \mathbf{w}_k^{\mathcal{T}} + \frac{v_k}{v_1} \Delta \mathbf{w}^t. \tag{36}$$

Recalling that $\widetilde{\mathbf{w}}_r^t = \sum_{k=1}^{K_r} v_k \mathbf{w}_k^t$, we rewrite $\widetilde{\mathbf{w}}_r^t$ as

$$\widetilde{\mathbf{w}}_r^t = \sum_{k=1}^{K_r} v_k \mathbf{w}_k^{\mathcal{T}} + \left( \sum_{k=1}^{K_r} \frac{v_k^2}{v_1} \right) \Delta \mathbf{w}^t \tag{37}$$

Noting that the norm of the first term in the right side of eq. (37) is finite and recalling that $\|\widetilde{\mathbf{w}}_r^t\| = +\infty$, we have, $\|\Delta \mathbf{w}^t\| = +\infty$, which, in conjunction with eq. (36), implies that $\|\mathbf{w}_k^t\| = +\infty$ for all $1 \leq k \leq K_r$.

Next, let us look at the update of $\mathbf{w}_1$ and $\mathbf{w}_2$. If there exist two elements in $\mathcal{V}_r$ that have different signs (without loss of generality, we assume $v_1 > 0$ and $v_2 < 0$), then

$$\mathbf{w}_1^{t+1} = \mathbf{w}_1^t - \eta_t \nabla_{\mathbf{w}_1^t} \ell(\mathbf{W}), \tag{38}$$

$$\mathbf{w}_2^{t+1} = \mathbf{w}_2^t - \eta_t \nabla_{\mathbf{w}_2^t} \ell(\mathbf{W}) = \mathbf{w}_1^t + \eta_t \left| \frac{v_1}{v_2} \right| \nabla_{\mathbf{w}_1^t} \ell(\mathbf{W}). \tag{39}$$

which can be rewriten as

$$\mathbf{w}_1^t = \mathbf{w}_1^{\mathcal{T}} - \sum_{s=\mathcal{T}+1}^{t-1} \eta_t \nabla_{\mathbf{w}_1^t} \ell(\mathbf{W}) = \mathbf{w}_1^{\mathcal{T}} + \Delta \mathbf{w}^t,$$

$$\mathbf{w}_2^t = \mathbf{w}_2^{\mathcal{T}} - \left| \frac{v_1}{v_2} \right| \Delta \mathbf{w}^t.$$

Recalling that $\|\Delta \mathbf{w}^t\| = +\infty$ as $t \to +\infty$ and noting that the first two neurons are activated after $\mathcal{T}$ iterations, we have, for $\forall \mathbf{x}_i \in \mathcal{B}_r$ and $t > \mathcal{T}$, the following two inequalities always hold

$$
\begin{aligned}
\mathbf{w}_1^{t\mathsf{T}} \mathbf{x}_i &= \mathbf{w}_1^{\mathcal{T}\mathsf{T}} \mathbf{x}_i + \Delta \mathbf{w}^{t\mathsf{T}} \mathbf{x}_i > 0, \\
\mathbf{w}_2^{t\mathsf{T}} \mathbf{x}_i &= \mathbf{w}_2^{\mathcal{T}\mathsf{T}} \mathbf{x}_i - \frac{v_1}{v_2} \Delta \mathbf{w}^{t\mathsf{T}} \mathbf{x}_i > 0.
\end{aligned}
\tag{40}
$$

Since $\Delta \mathbf{w}^t$ belongs to the space spanned by the samples in $\mathcal{B}_r$, $\Delta \mathbf{w}^t$ cannot be perpendicular to all $\mathbf{x}_i \in \mathcal{B}_r$. Thus, when $t \to +\infty$, we can find a sample $\mathbf{x}_r \in \mathcal{B}_r$ such that $\|\Delta \mathbf{w}^{t\mathsf{T}} \mathbf{x}_r\| = +\infty$. If $\Delta \mathbf{w}^{t\mathsf{T}} \mathbf{x}_r > 0$, then we have $\mathbf{w}_2^{t\mathsf{T}} \mathbf{x}_r < 0$, which contradicts eq. (40). If $\Delta \mathbf{w}^{t\mathsf{T}} \mathbf{x}_r < 0$, then $\mathbf{w}_1^{t\mathsf{T}} \mathbf{x}_r < 0$, which also contradicts eq. (40). As a result, all elements in $\mathcal{V}_r$ have the same sign.

Next, we prove that all samples in the same pattern partition have the same label. First consider the case when all elements in $\mathcal{V}_r$ are positive. If there exists a sample $\mathbf{x}_{sp} \in \mathcal{B}_r$ such that $y_{sp} = -1$ when $t = +\infty$, then

$$
\mathcal{L}(\mathbf{W}) = \frac{1}{n} \sum_{i=1}^n \exp(-y_i f(\mathbf{x}_i)) > \frac{1}{n} \exp(-y_{sp} \widetilde{\mathbf{w}}_r^{t\mathsf{T}} \mathbf{x}_{sp}) = \frac{1}{n} \exp(\sum_{k=1}^{K_r} v_k \mathbf{w}_k^{t\mathsf{T}} \mathbf{x}_{sp}) > \frac{1}{n},
$$

which contradicts that $\mathcal{L}(\mathbf{W}) < 1/n$.

Next, consider the case when all elements in $\mathcal{V}_r$ are negative. If there exists a sample $\mathbf{x}_{sp} \in \mathcal{B}_r$ such that $y_{sp} = +1$, we have

$$
\mathcal{L}(\mathbf{W}) = \frac{1}{n} \sum_{i=1}^n \exp(-y_i f(\mathbf{x}_i)) > \frac{1}{n} \exp(-y_{sp} \widetilde{\mathbf{w}}_r^{t\mathsf{T}} \mathbf{x}_{sp}) = \frac{1}{n} \exp(-\sum_{k=1}^{K_r} v_k \mathbf{w}_k^{t\mathsf{T}} \mathbf{x}_{sp}) > \frac{1}{n},
$$

which also leads a contradiction. Combining these two results, we conclude that if all elements in $\mathcal{V}_r$ are positive, then all samples in $\mathcal{B}_r$ have label $+1$, and if all elements in $\mathcal{V}_r$ are negative, then all samples in $\mathcal{B}_r$ have label $-1$.

Finally, note that $\widetilde{\mathbf{w}}_r$ is updated by

$$
\widetilde{\mathbf{w}}_r^{t+1} = \widetilde{\mathbf{w}}_r^t + \eta_t \sum_{k=1}^{K_r} v_k \nabla_{\mathbf{w}_k^t} \ell(\mathbf{W}) = \widetilde{\mathbf{w}}_r^t + \eta_t \big( \sum_{k=1}^{K_r} v_k^2 \big) \exp(-y_i \widetilde{\mathbf{w}}_r^{t\mathsf{T}} \mathbf{x}_i) y_i \mathbf{x}_i,
$$

which can be rewritten as

$$
\widetilde{\mathbf{w}}_r^{t+1} = \widetilde{\mathbf{w}}_r^t + \eta_t \big( \sum_{k=1}^{K_r} v_k^2 \big) \nabla \ell_r(\widetilde{\mathbf{w}}_r^t).
\tag{41}
$$

Applying Theorem 3.4 to eq. (41) with stepsize $\hat{\eta}_t = \eta_t \big( \sum_{k=1}^{K_r} v_k^2 \big)^{-1}$, we obtain that $\check{\mathbf{w}}_r^t$ converges in the direction of the max-margin classifier over all samples in $\mathcal{B}_r$, i.e.,

$$
\left\| \frac{\mathbb{E} \check{\mathbf{w}}_r^t}{\|\mathbb{E} \check{\mathbf{w}}_r^t\|} - \widehat{\mathbf{w}}_r \right\|^2 = \mathcal{O}\left( \frac{1}{\ln t} \right).
$$

