# OpenReview forum: "When Will Gradient Methods Converge to Max-margin Classifier under ReLU Models?"
_ICLR.cc/2019/Conference_

### Official Review · AnonReviewer3 · 2018-10-31

**Rating:** 5
**Confidence:** 4

**Review:**

This paper studies ReLU model, or equivalently, one-layer-one-neuron model, for the classification problem. This paper shows if the data is linearly separable, gradient descent may converge to either a global minimum or a sub-optimal local minimum, or diverges. This paper further studies the implicit bias induced by GD and SGD and shows if they converge, they can have a maximum margin solution.

Comments:
1. Using ReLU model for linearly separable data doesn't make sense to me. When ReLU is used, I expect some more complicated separable condition.
2. This paper only studies one-layer-one-neuron model, which is a very restricted setting. It's hard to see how this result can be generalized to the multiple-neuron case.
3. The analysis follows closely with previous work in studying the implicit bias for linear models.

---

### Official Review · AnonReviewer1 · 2018-11-01
**Importance of ReLU networks and max-margin used in this paper are unclear.**

**Rating:** 4
**Confidence:** 3

**Review:**

Recently, the implicit bias where gradient descent converges the max-margin classifier was shown for linear models without an explicit regularization.
This paper tries to extend this result to ReLU network, which is more challenging because of the non-convexity.
Moreover, a similar property of stochastic gradient descent is also discussed.

The implicit bias is a key property to ensure the superior performance of over-parameterized models, hence this line of research is also important.
However, I think there are several concerns as summarized below.

1. I'm not sure about the significance of the ReLU model (P) considered in the paper.
Indeed, the problem (P) is challenging, but an obtained model is linear defined by $w$.
Therefore, an advantage of this model over linear models is unclear.

Moreover, since the max-margin in this paper is defined by using part of dataset and it is different from the conventional max-margin, the generalization guarantees are not ensured by the margin theory.
Therefore, I cannot figure out the importance of an implicit bias in this setting (, which ensures the convergence to this modified max-margin solution).
In addition, the definition of the max-margin seems to be incorrect: argmin max -> argmax min.

2. Proposition 1 (variance bound) gives a bound on the sum of norms of stochastic gradients.
However, I think this bound is obvious because stochastic gradients of the ReLU model (P) are uniformly bounded by the ReLU activation.
Combining this boundedness and decreasing learning rates, the bound in Proposition 1 can be obtained immediately.
Moreover, the validity of an assumption on $w_t$ made in the proposition should be discussed.

3. Lemma F.2 is key to show the main theorem, but I wonder whether this lemma is correct.
I think the third equation in the proof seems to be incorrect.

---

### Official Review · AnonReviewer2 · 2018-11-03
**A theoretical paper with very stringent assumptions.**

**Rating:** 5
**Confidence:** 5

**Review:**

This paper considers the binary classification problem with exponential loss and ReLu activation function (single neuron). The authors characterize the asymptotic loss landscape by three different types of critical points. They prove that gradient descent (GD) will result in four different regions and provide convergence rates for GD to converge to an asymptotic global minimum, asymptotic local minimum and local minimum under certain assumptions. The authors also provide convergence results for stochastic gradient descent (SGD) and provide extensions to leaky ReLu activation and multi-neuron networks. The paper is well written and the results are mostly clearly presented. This paper mostly follows the line of research by Soudry et al. (2017, 2018), while it has its own merit due to the ReLu activation function considered. However, there are many strong assumptions that are not carefully verified and I really have concerns about the contribution of this paper since they simplify their analysis and results merely by imposing stringent conditions. In particular, I have the following major comments about the paper:

1.	In the definition of max-margin direction, why you use \argmin_{w} max_{i} (w^{\top}x_i)? It seems to me that the definition should be \argmax_{w} min_{i} (w^{\top}x_i). This definition keeps appearing in multiple places in the main paper.
2.	In the proof of Theorem 3.2, I am confused by the argument of the case that \hat w^{+} is not in the linearly separable region. More clarification is needed to make the proof rigorous.
3.	In the analysis of Theorem 3.3 and 3.4, the authors make a very stringent assumption that the iterate w_t staying in linear separable region for all t>\mathcal{T}. This assumption seems too strong, which should be verified rather than imposed in analysis of SGD. Note that even the example shown in Proposition 2 is still very restrictive (you require all the positive examples or negative examples are very close to one another).
4.	Furthermore, in the analysis of SGD, the authors did not specify the assumption that \hat w^{+} lies in the linear separable region, which is also required in this theorem and also very strong. Given such strong assumptions, the analytic results seem to be trivial and it is hard to evaluate the authors’ contribution.
5.	For the convergence results of SGD, the current rate is derived on the distance between \|E[w_t] - \hat{w}\|^2. Can you provide similar results for mean square error (E\| w_t - \hat{w} \|^2)?
6.	In multi-neuron case, the authors again make very strong assumptions that all the neurons have unchanging activation status. This is not easily achievable without careful characterization or other rigorous assumptions. Under such strong assumptions, the extension to multi-neuron again seems not very meaningful.

Other minor comments:
1.	The references are not correctly cited. For instance, please correct the use of parenthesis in “… which is different from that in (Soudry et al., 2017, Corollary 8)” and “… hold for various other types of gradient-based algorithms Gunasekar et al. (2018)”.
2.	The sentence “…, which the nature of convergence is different from …” does not read well. Should it be “where” or “of which”?

---

### Meta-Review · Area_Chair1 · 2018-12-08

**Confidence:** 4
**Recommendation:** Reject

**Metareview:**

The reviewers and AC note the following potential weaknesses: 1) the proof techniques largley follow from previous work on linear models 2) it’s not clear how signficant it is to analyze a one-neuron ReLU model for linearly separable data.